# A Perturbation Analysis of Input Transformations for Adversarial Attacks

## Abstract

Many defenses for Convolutional Neural Networks are based on a simple observation that the adversarial examples are not robust and small perturbations to the attacking input often recover the desired prediction. Intuitively, the adversarial examples occupy a very small cone in the decision space which is surrounded by a large area corresponding to the correct class. While the intuition is simple, a detailed understanding of this phenomenon is missing from the research literature.

We identify a family of defense techniques that are based on the instability assumption. The defenses include deterministic lossy compression algorithms and randomized perturbations to the input that all lead to similar gains in robustness. We present a comprehensive experimental analysis of when and why perturbation defenses work and potential mechanisms that could explain their effectiveness (or ineffectiveness) in different settings.

## 1 Introduction

The attacks on Convolutional Neural Networks, such as Carlini & Wanger (Carlini & Wagner, 2017) or PGD (Madry et al., 2017), generate strategically placed modifications that can be easily dominated by different types of perturbations resulting in correct predictions (Dziugaite et al., 2016; Roth et al., 2019). This suggests that the standard adversarial examples are not robust. Many defense techniques explicitly leverage this property and can be retrospectively interpreted as perturbations of the input images. However, a detailed understanding of this phenomenon is lacking from the research literature including: (1) what types of perturbations work and what is their underlying mechanism, (2) whether all attacks exhibit this property, and (3) possible counter-measures attackers can employ to defeat perturbation defenses.

We can interpret a large number of recent defenses as a type of input perturbations, for example, feature squeezing (Xu et al., 2017), frequency or JPEG compression (Dziugaite et al., 2016), randomized smoothing (Cohen et al., 2019), and perturbation of network structure or the inputs randomly (Jafarnia-Jahromi et al., 2018; Zhang & Liang, 2019; Guo et al., 2017). The defense techniques exhibit very similar gains in robustness. To show it, we start with a simple model where every example is passed through a lossy channel (stochastic or deterministic) prior to model inference. This channel induces a small perturbation to the input. We optimize the perturbation to be small enough as not to affect the prediction accuracy on clean examples but large enough to dominate any adversarial attack. We find that this trade-off is surprisingly consistent across very different families of input perturbations, where the relationship between channel distortion (the $L_2$ distance between channel input and output) and robustness is very similar.

Why are some state-of-the-art attacks are sensitive to perturbation-based defenses? We find that many attacks execute an optimization procedure that finds an adversarial image that is very close to the original image in terms of of $L_1$, $L_2$, or $L_\infty$ norm. The resultant optimum, i.e., the adversarial image, tends to exhibit a higher level of instability than natural examples, which we demonstrate from the perspective of a first-order and second-order analysis. By instability we mean that small perturbations of the example can affect the prediction confidences.

The unification of perturbation-based defense also gives us some insight into how an attacker might avoid them.

Our experiments suggest that all the perturbation based defenses are vulnerable to the same types of attack strategies. We argue that the optimization procedure in the attacker should find the smallest distance from the original image that closes the recovery window. In fact, we can devise a generic attacker that attacks a particularly strong lossy channel, based on the additive Laplace noise, and adaptive attacks designed on this channel are often successful against other defenses. This result implies that for many input perturbation defenses the attacker need not be fully adaptive, i.e., they do not need to know exactly what kind of transformation is used to defend the network.

## 2 RELATED WORK

Much of the community's current understanding of adversarial sensitivity in neural networks is based on the seminal work by Szegedy et al. (2014). Multiple contemporaneous works also studied different aspects of this problem, postulating linearity and over-parametrization as possible explanations (Goodfellow et al., 2014; Biggio et al., 2013). Since the beginning of this line of work, the connection between compression and adversarial robustness has been recognized. The main defense strategies include: the idea of defensive network distillation[1] (Papernot et al., 2015), quantizing inputs using feature squeezing (Xu et al., 2017), the thermometer encoding as another form of quantization (Buckman et al., 2018), JPEG compression harnessed by (Dziugaite et al., 2016; Guo et al., 2017; Das et al., 2017; 2018; Aydemir et al., 2018; Liu et al., 2019). Other line of research leveraged connection between randomization and adversarial robustness: Pixel Deflection (Prakash et al., 2018), random resizing and padding of the input image (Xie et al., 2017), and total variance minimization (Guo et al., 2017). In our work we unify the methods based on compression and randomization that are applied to the input images.

While all of the aforementioned defenses were later broken (Carlini & Wagner, 2017; Athalye et al., 2018), it is important to understand why these approaches afforded any form of robustness. The community actually lacks consensus on this point: Szegedy et al. (2014) suggest that neural networks have *blind spots*, Xu et al. (2017) suggest that quantization makes the adversarial search space smaller, Buckman et al. (2018) suggest the linearity is the main culprit and quantized inputs break up linearity.

Zhang & Liang (2019) inject random Gaussian noise into an image and then discretize it. This method fits into the noisy channel framework with different definitions of $C(x)$. They show improved performance of this combined model in the non-adaptive setting. Our experiments find that simply injecting Gaussian noise (or even Uniform or Laplace noise) is an equally effective defense method. We also show that the discretization is not essential to good performance if the level of noise is appropriately tuned. The imprecise channel defense in a neural network is also related to the idea of gradient masking or gradient obfuscation, i.e., a hard to differentiate layer (Papernot et al., 2017). In this work, the backward pass computation is perturbed to make it difficult for a gradient-based attack to synthesize an adversarial image (while the forward pass is kept the same). We implement both non-adaptive attacks and adaptive that can observe the channel and take an approximate gradient through it. We further focus our study on the families of white-box attacks proposed by Carlini & Wagner (2017), and their adaptive variants.

Noise injection can be much more powerful than regularization or a dataset augmentation method. The dropout algorithm can be seen as applying noise to the hidden units. The dropout randomization (Feinman et al., 2017) was used to create a defense that was not completely broken and required a high distortion added to the adversarial examples (Carlini & Wagner, 2017). Many new defenses propose randomization through noise injection without considering the adversarial training (Zhang & Liang, 2019; Cohen et al., 2019). The work on injection of noise into inputs and each of the layers of neural networks by Liu et al. (2018) is a strong heuristic that led to defenses with theoretical guarantees. The random smoothing provides a certified robustness up to a certain threshold of input distortion (and is not designed to be robust beyond the threshold) by utilizing inequalities from the differential privacy literature (Lecuyer et al., 2018). Cohen et al. (2019) improve the theoretical bounds of methods that randomly smooth the input examples. Recent work focuses on combination of randomized smoothing with adversarial training and achieves state of the art in terms of the provable robustness (Salman et al., 2019).

---

[1]The distillation is a form of compression, however, the defensive distillation does not result in smaller models.

## 3 LOSSY CHANNEL MODEL

We consider convolutional neural networks that take $w \times h$ (width times height) RGB digital images as input, giving an example space of $\mathcal{X} \in (255)^{w \times h \times 3}$, where $(z)$ denotes the integer numbers from 0 to $z$. We consider a discrete label space of $k$ classes represented as a confidence value $\mathcal{Y} \in [0, 1]^k$. Neural networks are parametrized functions (by a weight vector $\theta$) between the example and label spaces $f(x; \theta) : \mathcal{X} \mapsto \mathcal{Y}$.

An adversarial input $x_{adv}$ is a perturbation of a correctly predicted example $x$ that is incorrectly predicted by $f$.

$$f(x) \neq f(x_{adv})$$

The *distortion* is the $\ell_2$ error between the original example and the adversarial one:

$$\delta_{adv} = \|x - x_{adv}\|_2$$

### 3.1 MODEL

Approximating $f(\cdot)$ with a less precise version $\bar{f}(\cdot)$ can counter-intuitively make it more robust (Dziugaite et al., 2016):

$$f(x) = \bar{f}(x_{adv})$$

Intuitively, a lossy version of $f$ introduces noise into a prediction which dominates the strategic perturbations found by an adversarial attack procedure. It turns out that we can characterize a number of popular defense methodologies with this basic framework.

Let $x$ be an example and $f$ be a trained neural network. Precise evaluation means running $f(x)$ and observing the predicted label. Imprecise evaluation involves first transforming $x$ through a deterministic or stochastic noise process $C(x) = C[x' \mid x]$, and then evaluating the neural network

$$y = f(x') \quad x' \sim C(x)$$

We can think of $C(x)$ as a noisy channel (as in signal processing). The *distortion* of a $C(x)$ is the expected $\ell_2$ reconstruction error:

$$\delta_c = \mathbf{E}[\|C(x) - x\|_2],$$

which is a measure of how much information is lost passing the example through a channel.

This paper shows that there is a subtle trade-off between $\delta_c$ and $\delta_{adv}$. In particular, we can find $\delta_c$ such that $\delta_c >> \delta_{adv}$ and $f(x) = f(C(x_{adv}))$. We show that compression and randomization based techniques exhibit this property.

EXAMPLE OF DETERMINISTIC CHANNEL

When $C(x)$ is deterministic it can be thought of as a lossy compression technique. Essentially, we run the following operation on each input example:

$$x' = \texttt{compress}(x)$$

One form of compression for CNNs is color-depth compression. Most common image classification neural network architectures convert the integer valued inputs into floating point numbers. We abstract this process with the $\texttt{norm}$ function that for each pixel $n \in (255)$ maps it to a real number $v \in [0, 1]$ by normalizing the value and the corresponding $\texttt{denorm}$ function that retrieves the original integer value (where $\lfloor \rceil$ denotes the nearest integer function) [2]:

$$\texttt{norm}(n) := \frac{n}{255} \qquad \texttt{denorm}(v) := \lfloor 255 * v \rceil$$

This process is normally reversible $v = \texttt{norm}(\texttt{denorm}(v))$, but we can artificially make this process lossy. Consider a parametrized $C(\cdot)$ version of the color-depth compression function:

$$C(v, b) := \frac{1}{2^b - 1} \cdot \lfloor (2^b - 1) * v \rceil$$

By decreasing $b$ by $\Delta b$ we reduce the fidelity of representing $v$ by a factor of $2^{\Delta b}$ (for the $b$ bits of precision).

---

[2]More complex normalization schemes exist but for ease of exposition we focus on this simple process.

EXAMPLE OF STOCHASTIC PERTURBATION

The channel model is particularly interesting when $C(x)$ is stochastic. Randomization has also been noted to play a big role in strong defenses in prior work (Madry et al., 2017; Zhang & Liang, 2019; Cohen et al., 2019). For example, we could add independent random noise to each input pixel:

$$x' = x + \epsilon$$

We consider two schemes, Gaussian $\epsilon \sim N(0, \sigma)$ and additive Uniform noise $\epsilon \sim U(-B, B)$ which add independent noise to each pixel.

One of the advantages of randomization is that an adversary cannot anticipate how the particular channel $C$ will transform an input before prediction. However, there is another subtle advantage to randomization. Randomized approaches can partially recover their loss in accuracy due to imprecision by averaging over multiple instances of the perturbation. In classification problems, we can take the most frequent label seen after $T$ perturbation trials:

$$\bar{f}(x) = \arg \max_{1...k} \sum_i^T f(x + \epsilon)$$

## 3.2 PERTURBATION ANALYSIS

While the intuition is that the channel's perturbations dominate strategically placed distortions in an adversarial example, the underlying mathematical mechanism of why recovery is possible is not clear. We start with the hypothesis that synthesized adversarial examples have *unstable* predictions–meaning that small perturbations to the input space can change confidence values drastically. How do we quantify instability?

Let $f(x)$ be a function that maps an image to a single class confidence value (i.e., a scalar output). We want to understand how $f(x)$ changes if $x$ is perturbed by $\epsilon$. We can apply a Taylor expansion of $f$ around the given example $x$:

$$f(x + \epsilon) \approx f(x) + \epsilon^T \nabla_x f(x) + \frac{1}{2} \epsilon^T \nabla_x^2 f(x) \epsilon + ...$$

where $\nabla_x f(x)$ denotes the gradient of the function $f$ with respect to $x$ and $\nabla_x^2 f(x)$ denotes the Hessian of the function $f$ with respect to $x$. The magnitude of the change in confidence is governed by the Taylor series terms in factorially decreasing importance. $\|\epsilon\|_2$ is exactly the distortion measure $\delta_c$ described at the beginning of Section 3.1. Thus, the expression is bounded in terms of the *operator* norm, or the maximal change in norm that could be induced, of each of the terms (see Appendix A):

$$\epsilon^T \nabla_x f(x) + \frac{1}{2} \epsilon^T \nabla_x^2 f(x) \epsilon + ... \leq \delta_c M_1(x) + \frac{1}{2} \delta_c^2 M_2(x) + ...$$

As $\nabla_x f(x)$ is a vector, this is simply the familiar $\ell_2$ norm, and for the second order term this is the maximal eigenvalue:

$$M_1(x) = \|\nabla_x f(x)\|_2 \quad M_2(x) = \lambda_{max}(\nabla_x^2 f(x))$$

When $M_1$ and $M_2$ are larger this means there is a greater propensity to change the prediction for small perturbations. We will show experimentally that for certain types of attacks the $M_1$ and $M_2$ values around adversarial examples exhibit signs of instability compared to those around natural examples–suggesting a mathematical mechanism of why recovery is possible.

## 4 EXPERIMENTS

Our experiments evaluate the efficacy of imprecision based defenses in a number of different adversarial problem settings.

### 4.1 EXPERIMENTAL SETUP

We run our experiments using ResNet-18 on CIFAR-10 and ResNet-50 on ImageNet dataset using P-100 GPUs (16GB memory). We explore a number of different attacks that are implemented in

the foolbox library (Rauber et al., 2017). In each experiment we measure the test accuracy (%), the confidence of predictions, and distances between the original images and either their adversarial counterparts or the recovered images after applying one of the defenses. We present our results for *non-targeted attacks*; if the adversary is successful it induces any misclassification. We experiment with many gradient-based attacks provided in the foolbox library that explore different optimization algorithms and distance measures, for instance:

- **LBFGS** minimizes the distance between the input image and the adversarial example as well as the cross-entropy between the predictions for the adversarial and the input image; introduced by Szegedy et al. (2014) and further extended in Tabacof & Valle (2015).

- **Carlini-Wagner** $L_2$ (C&W $L_2$) is a generalization of the LBFGS attack that is devised after exhaustive search over possible space of: norms, loss functions, box optimization procedures, etc. (Carlini & Wagner, 2017).

- **BIM** $L_1$ is a modified version of the Basic Iterative Method that minimizes the $L_1$ distance (Kurakin et al., 2016).

- **FGSM** adds the sign of the gradient to the image, gradually increasing the magnitude until the image is misclassified Goodfellow et al. (2014).

- **PGD** $L_\infty$ the Projected Gradient Descent Attack that is an iterative version of the FGSM attack; we use the version that minimizes the $L_\infty$ distance (Madry et al., 2017).

We extend the Carlini-Wagner $L_2$ attack in its adaptive version so that the gradients are not obfuscated. We approximate the gradients for the backward pass on the compression layers as an identity function, similarly to (He et al., 2017; Athalye et al., 2018). More details on the setup can be found in the Appendix, Section C.13

## 1. Perturbation Defenses Give Similar Gains In Robustness

In the first experiment, we select 1000 random images from ImageNet and CIFAR-10 datasets. For each of these images we generate an adversarial attack with popular white-box gradient-based methods. Then, we run the adversarial images through different channels: Frequency Compression (FC), Color Depth reduction (CD), Uniform noise (Unif), Gaussian Noise (Gauss), SVD-based compression (SVD), Identity (Iden). The identity channel just passes through the adversarial image with no modification. We measure the accuracy of $f(C(x))$, which indicates the ability of the imprecise channels to recover the original label. We present the results in Figure 1 and also in the Supplement in Figure 7 (for all images in the test CIFAR-10 and dev ImageNet sets) and in Table 3 (for different channel parameters and five attacks).

When there is an identity channel, the adversarial attack is always successful. However, each of the imprecise channels is able to recover a substantial portion of original labels from the adversarial examples. It is important to note that we are evaluating these attacks in the setting, where any mis-classification is considered a success and the adversary is not aware of the defense.

**Importantly, all the channels can be tuned to recover very similar maximum accuracy after attacks.** For example, all the channels can achieve about 85% accuracy for CIFAR-10 after non-adaptive PGD or C&W attacks. This suggests that any form of imprecision with the right error magnitude is effective at defending against these types of attacks. We observe that the attacks that incur higher distortions such as FGSM or LBFGS decrease the accuracy more than the iterative attacks such as C&W $L_2$ or PGD $L_\infty$. This is because the iterative attacks find the adversarial images that are closer to the original images in terms of the corresponding distance measure that they optimize for in the input space. *The key is to ensure that the error introduced by the imprecise channels is big enough to dominate the adversarial perturbations but small enough to generate valid predictions.* Figure 1 illustrates this relationship (see also a detailed analysis for a single image presented in Figure 9 in the Appendix). For five different imprecise channels, we plot the channel distortion against the accuracy for CIFAR-10 and ImageNet datasets. We analyze FC, CD, Unif, Gauss, and SVD channels. The curves are qualitatively similar in the *low* noise regime, but they show more differences when higher distortions are incurred. The lowest accuracy across the datasets for high distortions is observed for content-preserving FC and SVD compression channels. The Gaussian and Uniform channels have very similar trends. They outperform other channels on ImageNet for

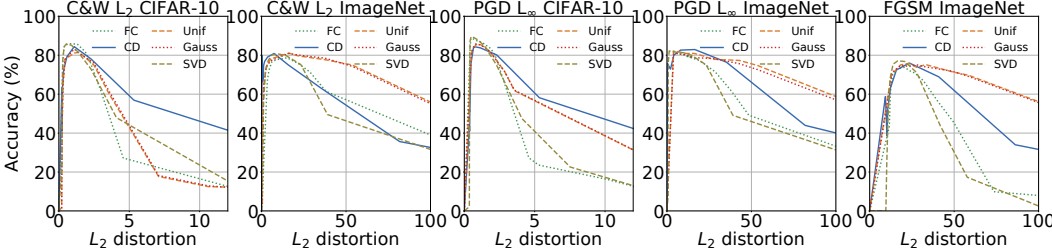

Figure 1: We plot the *channel distortion* against test accuracy (%). The distortion of the imprecise channels has to be large enough to recover the correct label but not so large that it degrades model performance. The base test accuracy is about 93.5% for CIFAR-10 and 83.5% for ImageNet on 1000 randomly chosen images (results for the full test CIFAR-10 set and the full dev ImageNet set can be found in the Appendix in Figure 7). The experiment is run for the C&W $L_2$ attack with 100 iterations and the PGD attack with 40 iterations.

Table 1: Transferability of the adversarial images created against a given noisy channel denoted as $A$ (adaptive attack specified in the first column) to the defense protected with a noisy channel denoted as $D$ (the defense with a noisy channel specified in the first row). Each result represents a recovery (%) of the adversarial examples (generated for $A$) to correct labels after applying the defense ($D$). We use 30% FC compression, 50% SVD compression, 4 bit values in CD, 0.03 noise level for Gauss and Laplace, and 0.04 noise level for the Uniform channel. We use 2000 images from the CIFAR-10 test set and 100 attack iterations with 5 binary steps to find the $c$ value (with initial $c$ value set to 0.01) for the adaptive C&W $L_2$ attack. The baseline test accuracy is 93.56%. The test accuracy of the noisy channels on clean images is given in the first row denoted: *Empty* (an empty attack).

| $A$ \ $D$ | FC | CD | SVD | Gauss | Uniform | Laplace |
|---|---|---|---|---|---|---|
| *Empty* | 93.32 | 93.01 | 93.12 | 92.53 | 91.6 | 91.35 |
| FC | **0.20** | 80.75 | 83.05 | 81.15 | 79.65 | 78.70 |
| CD | 3.85 | **0.70** | 43.60 | 47.30 | 60.45 | 62.35 |
| SVD | 1.99 | 47.96 | **0.77** | 46.52 | 62.87 | 65.75 |
| Gauss | 4.45 | 48.70 | 44.80 | 51.50 | 61.75 | 60.15 |
| Uniform | 3.45 | 30.30 | 30.60 | 30.15 | 48.05 | 51.55 |
| Laplace | 3.05 | 23.35 | 24.60 | **23.80** | **39.15** | **46.70** |

large distortions but are less performant on low-resolution CIFAR-10 images, where the CD channel achieves higher accuracy.

## 2. ATTACKS ARE TRANSFERABLE

Many input transformation defenses are broken. If the attacker has full knowledge of the defense, it is possible to construct an attack that is impervious to the defense. This is called the *adaptive* setting. Since the underlying mechanisms of input transformations are similar, we find that an attacker *does not need to be fully adaptive*. The attacker can assume a particular strong defense and that same adversarial input often transfers to other defenses. We narrow the attacker to a single adaptive step (for details see Section C.8 in the Appendix). Even in this weak adaptive setting, the deterministic channels are fully broken but the randomized channels retain relatively high accuracy above 23.8%. We show in the Appendix in Figure 14 that the randomized defenses can also be broken when the adversary is given an unlimited number of adaptive steps.

**Laplace attacked images transfer the best to other defenses**. They decrease the accuracy of the defense models by at least 44.3% (for Laplace itself). Table 1 shows that FC attacked images do not transfer well to other defenses; the maximum drop in accuracy of the model protected by other defenses is 12.26%. Most adversarial images (against a given defense) transfer very well to the FC defense, i.e. an adversarial image against any defense (e.g. CD, SVD, Gauss, Uniform, or Laplace) is also adversarial against the FC defense. The adversarial images generated against the Uniform

defense show better transfer to other defenses in comparison to the adversarial images generated against the Gaussian defense. This is because the higher noise level is applied in the Uniform defense. We observe analogous trends for the ImageNet dataset and present the results in the supplement (Tables 6 and 7).

### 3. Recoverable Range shrinks with higher attack distortion

We use an example from the ImageNet dataset, the ResNet-50 model, and set the stochastic channel to the Gaussian noise. We start from an adversarial example generated with the Carlini & Wagner (non-adaptive) $L_2$ attack and for consecutive subplots (in the left to right and top to bottom sequence), we increase the attack strength and incur higher distortion of the adversarial image from the original image. For a single plot, we increase the $L_2$ distance of the output from the channel to the adversarial example by increasing the Gaussian noise (controlled by parameter $\sigma$). For $L_2$ distances incurred by different noise levels, we execute 100 predictions. In Figure 9 in the Appendix, we use the frequency count and report how many times the model predicts the original, adversarial or other class. The plot shows what range of distances from the adversarial image reveal the correct class. For the adversarial examples that are very close to the original image (e.g. adversarial distance of 0.006 for the top-left figure), the window of recovery (which indicates which strengths of the random noise can recover the correct label) is relatively wide, however, as we increase the distance of the adversarial image from the original image, the window shrinks and finally we are not enable to recover the correct label.

For attacks that incur more input distortion or the stronger white-box attacks, we can resort to other statistics as proposed in Roth et al. (2019). They show that adversarial examples are much closer to the unperturbed sample than to any other neighbor and use the fact that the probability of the correct class increases faster than the probability of the highest other class when adding noise with a small to intermediate magnitude to the adversarial example.

### 4. Comparison with other methods

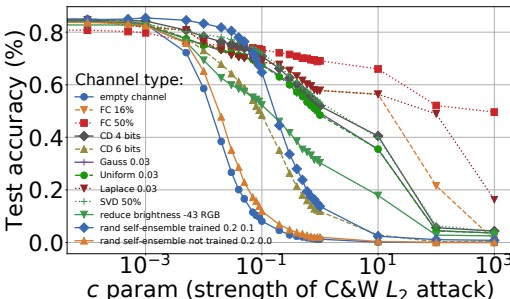 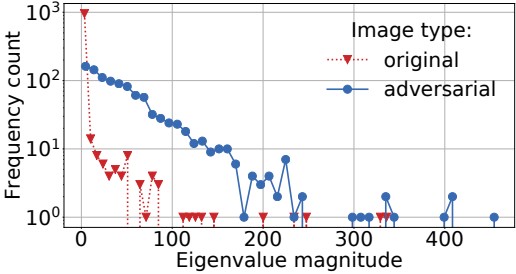

Figure 2: Non-adaptive attack and the recoverable ranges in terms of the c parameter that controls the strength of the C&W $L_2$ attack. We systematically change the c parameter and keep the parameters for the channels unchanged. We use the VGG-16 network and all images from the CIFAR-10 test set. The test accuracy of the model without any noise layers and on the clean data is 85.23%.

Figure 3: The top eigenvalues of the Hessians with respect to (w.r.t.) the input 1024 images from the CIFAR-10 dataset trained on the ResNet-18 architecture. We plot the histogram that shows counts of magnitudes for the eigenvalues. Yao et al. (2018) show analysis of the Hessian w.r.t. parameters and we extend it to analyze Hessian w.r.t. inputs.

In Figure 2 we present the accuracy of different channels as the strength of the C&W attack is systematically increased. The strength is controlled by the $c$ tradeoff-parameter that is used to set the relative importance of distance and confidence.

We present a related approach which is the RSE (Random Self-Ensemble) network with 0.2 noise level in the first layer and 0.1 noise level in the remaining layers (as recommended in Liu et al. (2018)). This defense does better for lower distortion levels (c value below 0.1) than other noisy channels, but then its accuracy deterioration is faster for higher distortion levels. The Laplace channel gives the highest accuracy for high values of the $c$ parameter (above 1.0). The CD, FC, SVD, Gauss, and

Uniform channels show similar trends. We also include a very simple channel that reduces brightness of an image by subtracting an arbitrary value from each pixel. The comparison between very complex approaches and a simple input transformation is informative–as they largely follow the same trends.

## 5. ADVERSARIAL EXAMPLES ARE UNSTABLE

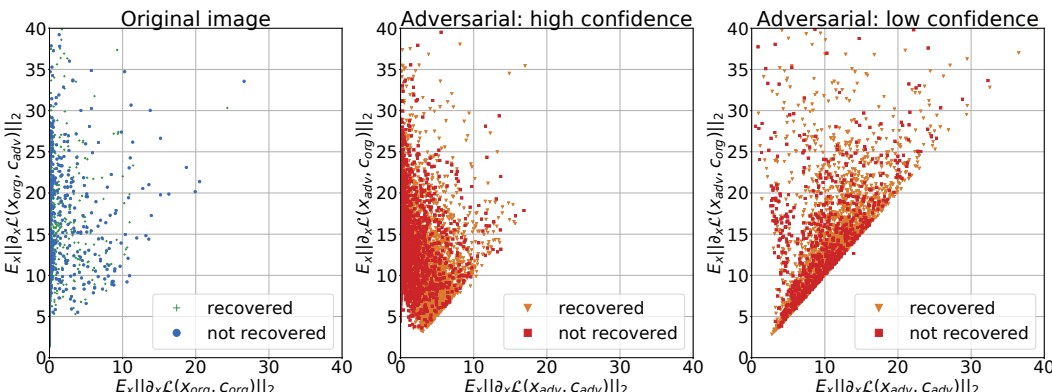

Figure 4: Comparison of magnitude of gradients for original and adversarial images.

**First-order analysis.** The key question is why adversarial examples are more sensitive to perturbations than natural inputs, when there is evidence that from an input perspective they are statistically indistinguishable. Our experiments suggest that this sensitivity arises from the optimization process that generates adversarial inputs. Based on the operator-norm analysis presented in Section 3.2, we plot in Figure 4 the $L_2$ norm of gradients w.r.t. the original $x_{org}$ and adversarial $x_{adv}$ images for original (correct) $c_{org}$ and adversarial classes $c_{adv}$. The work by Simon-Gabriel et al. (2019) also evaluates the norm of gradients of the network output with respect to its inputs. They show that the adversarial examples are primarily caused by large gradients of the classifier as captured via the induced loss. At first-order approximation in $\epsilon$, an $\epsilon$-sized $L_2$ *untargeted* adversarial attack increases the loss $\mathcal{L}$ at point $x$ by $\epsilon||\partial_x\mathcal{L}(x, c_{org})||_2$. Analogously, at first-order approximation in $\epsilon$, an $\epsilon$-sized $L_2$ *targeted* adversarial attack decreases the loss $\mathcal{L}$ at point $x$ by $\epsilon||\partial_x\mathcal{L}(x, c_{adv})||_2$. In our experiments, the recovered original images have lower norms of the gradients for both *original* and *adversarial* classes than the not recovered images. Additionally, the gradient w.r.t. original image for the original class is much smaller than the gradient w.r.t. adversarial image for the adversarial class. We also observe that the magnitudes of gradients change smoothly as we systematically add more Gaussian noise to the original or adversarial images (Figures 15, 16). For the adversarial image, the gradient w.r.t. adversarial image and for adversarial class decreases as we increase the attack strength and the confidence of adversarial predictions.

**Second-order analysis.** Our results in Figure 3 show that the adversarial inputs lead to noticeably higher Hessian spectrum than the original inputs. This suggests that the model predictions for the adversarial inputs are less stable than for the original images. Thus, perturbations of the adversarial images with some form of noise can easily change the classification outcome while the prediction for the original images are much more robust and do not lead to such unstable predictions.

## 5 CONCLUSION, LIMITATIONS, AND FUTURE WORK

The non-adaptive attacks are not robust since small changes to the adversarial input often recover the original label. This is an obvious corollary to the very existence of adversarial examples that by definition are relatively close to correctly predicted examples in the input space. Random perturbations of the input can dominate the strategically placed perturbations synthesized by an attack. In fact, the results are consistent across both deterministic and stochastic channels that degrade the fidelity of the input example. From the perspective of the attacker, the recovery window can be closed to make the perturbation based recovery techniques ineffective.

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

# A  PERTURBATION ANALYSIS: ADDENDUM

$$\epsilon^T \nabla_x f(x) + \frac{1}{2}\epsilon^T \nabla_x^2 f(x)\epsilon + ... \leq \delta_c\, M_1(x) + \frac{1}{2}\delta_c^2\, M_2(x) + ...$$

$$M_1(x) = \|\nabla_x f(x)\|_2 \quad M_2(x) = \lambda_{max}(\nabla_x^2 f(x))$$

$$(1) \quad \epsilon^T \nabla_x f(x) \leq \delta_c\, M_1(x)$$

From the Cauchy-Schwarz inequality: $\epsilon^T \nabla_x f(x) \leq \|\epsilon\|_2\, M_1(x)$

$$\|\epsilon\|_2\, M_1(x) = \delta_{adv}\, M_1(x) \leq \delta_c\, M_1(x) \ ( \text{ since } \delta_{adv} << \delta_c)$$

$$(2) \quad \nabla_x^2 f(x)\epsilon \leq \delta_c^2\, M_2(x)$$

From the definition of maximum eigenvalue : $\lambda_{max} \geq \dfrac{\epsilon^T \nabla_x^2 f(x)\epsilon}{\epsilon^T \epsilon}$

$$\epsilon^T \nabla_x^2 f(x)\epsilon \leq \|\epsilon\|_2^2 \lambda_{max} = \delta_{adv}^2 \lambda_{max} \leq \delta_c^2 \lambda_{max}$$

# B  COMPRESSION TECHNIQUES

## B.1  FFT-BASED COMPRESSION

We apply compression in the frequency domain to reduce the precision of the input images. Let $x$ be an input image, which has corresponding Fourier representation that re-indexes each tensor in the frequency domain:

$$F[\omega] = F(x[\mathbf{n}])$$

This Fourier representation can be efficiently computed with an FFT. The mapping is invertible $x = F^{-1}(F(x))$. Let $M_f[\omega]$ be a discrete indicator function defined as follows:

$$M_f[\omega] = \begin{cases} 1, \omega \leq f \\ 0, \omega > f \end{cases}$$

$M_f[\omega]$ is a mask that limits the $F[\omega]$ to a certain *band* of frequencies. $f$ represents *how much of the frequency domain* is considered. The *band-limited* spectrum is defined as, $F[\omega] \cdot M_f[\omega]$, and the band-limited filter is defined as:

$$x' = F^{-1}(F[\omega] \cdot M_f[\omega])$$

## B.2  SVD-BASED COMPRESSION

Analogously to the FFT-based method, we decompose an image with SVD transformation and reconstruct its compressed version with dominant singular values. The basis used in SVD are adaptive and determined by an image, as opposed to pre-selected basis used in FFT. This can result in higher quality for the same compression rate in case of SVD, however it is more computationally intensive than FFT-based compression.

# C  ADDITIONAL EXPERIMENTS FOR WHITE-BOX ATTACKS

## C.1  ACCURACY OF PERTURBATION DEFENSES ON CLEAN DATA

One pitfall of the imprecise channel defense is that it introduces errors whether or not there are any adversarial examples. The errors act as an upper-bound for the best possible test accuracy we can get under adversarial perturbations. For frequency compression, color depth compression, and uniform noise injection, we compare the test accuracy for different levels of imprecision. Table 2 shows the results for all test images from CIFAR-10 on the ResNet-18 architecture, for three of the imprecise channels, and for different noise settings.

Table 2: On CIFAR-10 with ResNet-18, we measure the max test accuracy for imprecise channels without any adversarial perturbation. This signifies the amount of accuracy we sacrifice with respect to the baseline test accuracy of the model (without any perturbations of the images): 93.56%.

| FC (%) | Acc. (%) | CD (bits) | Acc. (%) | Uniform ($\epsilon$) | Acc. (%) |
|---|---|---|---|---|---|
| 1 | 93.5 | 8 | 93.4 | 0.009 | 93.52 |
| 10 | 93.42 | 6 | 93.3 | 0.03 | 92.59 |
| 50 | 91.6 | 4 | 91.9 | 0.07 | 85.2 |
| 75 | 79.53 | 2 | 87.4 | 0.1 | 70.67 |

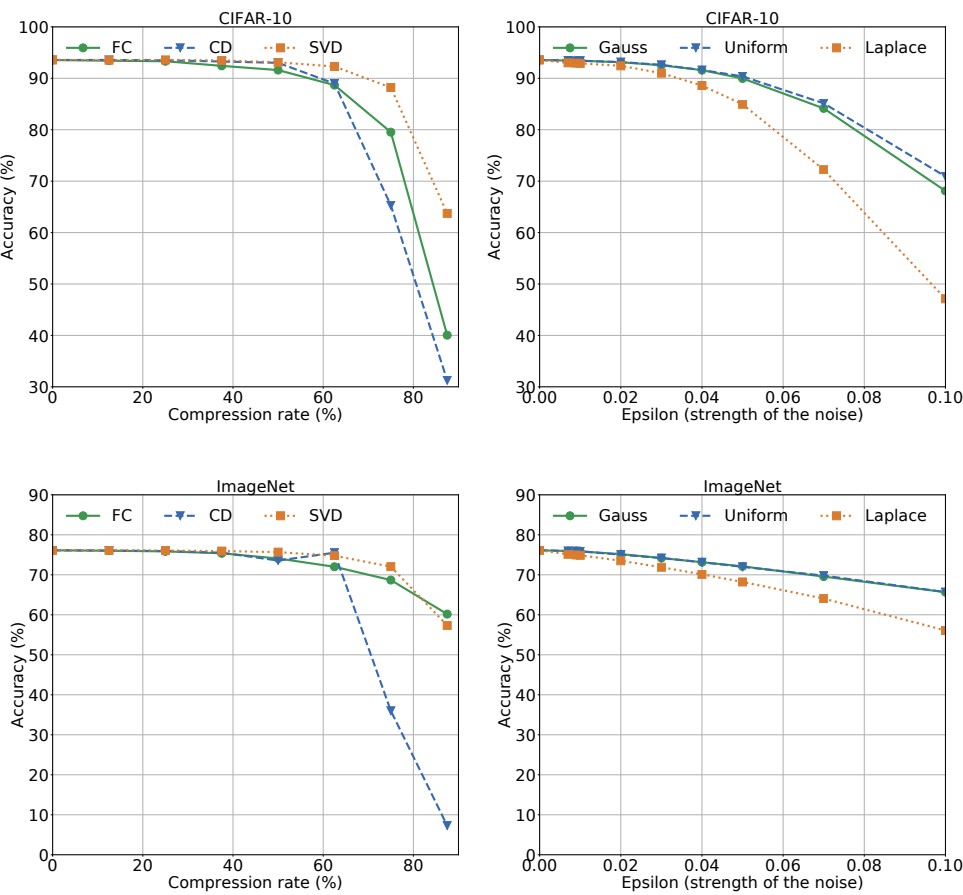

Figure 5: The test accuracy after passing clean images through six different noisy channels, where the added noise is controlled by the compression rate and epsilon parameters. We use full CIFAR-10 test set for ResNet-18, and full ImageNet validation set for ResNet-50.

The test accuracy of the models can be increased by training with compression, e.g., by using FFT based convolutions with 50% compression in the frequency domain increases the accuracy to 92.32%.

We present the results for six different noisy channels; three of them are compression based: FC, CD, SVD, and other three add different type of noise: Gauss, Uniform, and Laplace. For each of the compression based channels, we increase the compression rate systematically from 0 to about 90% (in case of the CD channel, the compression rate is computed based on how many bits are used per value). For the noise based channels, we increase the strength of the noise by controlling the epsilon parameter $\epsilon$ (in case of the Gaussian noise, it corresponds to the sigma parameter $\sigma$). The full result is presented in Figure 5.

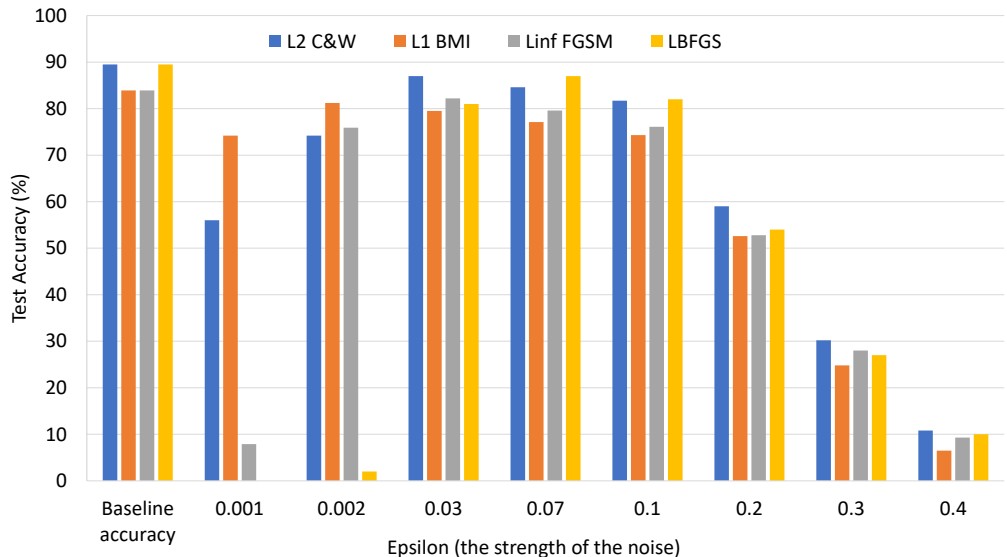

Figure 6: The test accuracy after four different attacks and the recovery via the uniform noise stochastic channel with different parameters $\epsilon$. For the Carlini-Wagner L2 attack, the best performing parameter $\epsilon = 0.03$. We run the experiment for ResNet-50 on ImageNet (1000 samples).

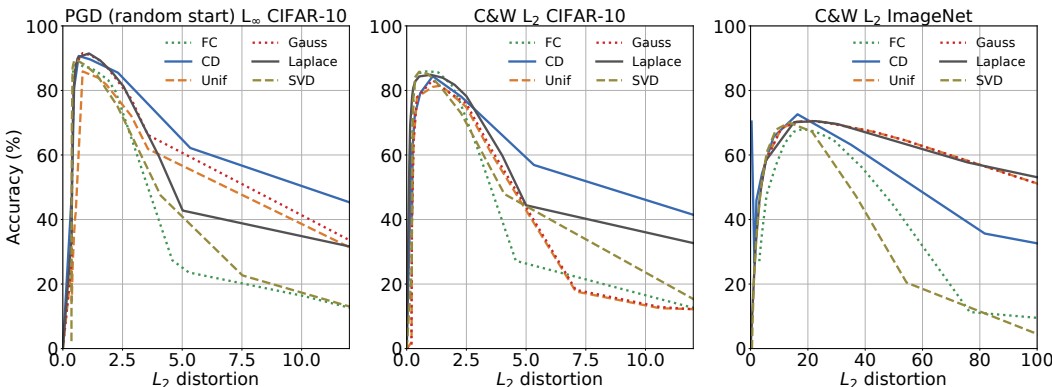

Figure 7: We plot the *channel distortion* against accuracy (%), analogously to Figure 1. The experiment is run for the PGD attack with 40 iterations and C&W $L_2$ attack with 100 iterations on all images from the test CIFAR-10 and the dev ImageNet datasets. The adversary is not aware of the defense. The test accuracy on the full clean data is 93.56% and 76.13% for CIFAR-10 and ImageNet, respectively. Interestingly enough, we observe that the C&W attack is on the sub-pixel level for ImageNet and rounding to the nearest 8 bit integers in the CD channel has high recovery rate, while for 7 bit integer the accuracy drops slightly due to imprecision.

Figure 6 presents the test accuracy (%) after four different attacks (3 different norms) and the recovery via the uniform noise stochastic channel in more detail.

## C.2 CHANNEL DISTORTION VS ACCURACY

We show channel distortion vs accuracy for the C&W $L_2$ attack on the full CIFAR-10 test set and full ImageNet dev set in Figure 7.

We further show the performance of the defenses across different attacks and 1000 images in Table 3.

Table 3: Given an adversarial input, we pass the input through a channel before prediction. We evaluate the accuracy (%) of the classifier over 1000 images, the best possible accuracy is listed in the Baseline column. For the channels, we report in parentheses: compression used (%), number of bits per value, and the strength of the attack ($\epsilon$). We use the attacks described in Section: 4.1

| Attack | Data set | Baseline | FC (%) | CD (bits) | Uniform ($\epsilon$) | Gauss ($\epsilon$) | Iden |
|---|---|---|---|---|---|---|---|
| BIM $L_1$ | CIFAR10 | 93.5 | 86.2 (20) | 85.1 (4) | 82.8 (0.03) | 82.1 (0.03) | 0 |
| LBFGS | CIFAR10 | 93.5 | 82.8 (50) | 80.2 (4) | 79.2 (0.04) | 79.3 (0.05) | 0 |
| C&W $L_2$ | CIFAR10 | 93.5 | 85.2 (20) | 84.4 (4) | 84.3 (0.01) | 84.8 (0.02) | 0 |
| FGSM | CIFAR10 | 93.5 | 79.0 (50) | 49.2 (4) | 49.4 (0.03) | 49.9 (0.03) | 0 |
| PGD $L_\infty$ | CIFAR10 | 93.5 | 88.6 (10) | 84.3 (5) | 85.7 (0.01) | 84.9 (0.02) | 0 |
| BIM $L_1$ | ImageNet | 83.5 | 81.5 (10) | 82.0 (4) | 81.2 (0.009) | 81.2 (0.009) | 0 |
| LBFGS | ImageNet | 83.5 | 71.7 (70) | 77.6 (4) | 76.4 (0.07) | 76.5 (0.07) | 0 |
| C&W $L_2$ | ImageNet | 83.5 | 78.7 (50) | 80.9 (4) | 81.4 (0.03) | 80.4 (0.03) | 0 |
| FGSM | ImageNet | 83.5 | 73.8 (50) | 76.0 (4) | 75.4 (0.03) | 75.4 (0.02) | 0 |
| PGD $L_\infty$ | ImageNet | 83.5 | 82.1 (5) | 82.9 (4) | 82.0 (0.007) | 80.9 (0.01) | 0 |

## C.3 CHANNEL ACCURACY ON CLEAN AND ADVERSARIAL EXAMPLES

We compare the accuracy of the perturbation channels on clean and adversarial examples in Figure 8. We plot the results for the whole test set from CIFAR-10 and the whole validation set from ImageNet. We use two deterministic channels (FFT compression denoted by FC and SVD compression) and two noisy channels (Gaussian and Uniform noise). We use C&W and PGD attacks. The *FC Clean* label denotes that we pass clean images through the channel that applies FFT compression. The *SVD C&W* label denotes that we pass adversarial images found with the C&W attack through the channel that applies SVD compression. We tune the channels for a given dataset across the attacks, and present the results in Table 4. The accuracy of the channels on clean data gives us the upper bound for the accuracy on the adversarial examples. Thus, we choose the channel parameter based on the highest accuracy on the adversarial images.

Table 4: **Channel tuning**. The best parameters for the perturbation channels when tuned on the PGD and C&W attacks.

| Channel \ Dataset | CIFAR-10 | ImageNet |
|---|---|---|
| FC (%) | 20 | 60 |
| SVD (%) | 40 | 70 |
| Gauss ($\epsilon$) | 0.015 | 0.04 |
| Uniform ($\epsilon$) | 0.025 | 0.04 |

## C.4 NEIGHBORHOOD OF ADVERSARIAL EXAMPLES

Please, see Figure 9 and description in Section 4.1.

## C.5 DISTRIBUTIONS OF DELTAS BETWEEN INPUTS & OUTPUTS FOR CHANNELS

We plot the distribution of *deltas* for six imprecise channels in Figure 10. We compute the *deltas* by subtracting an original image from the perturbed adversarial image and plot the histograms of differences. We use an image from the ImageNet dataset. For all the examples, the correct labels were recovered. We use the C&W attack with 1000 iterations and the initial value $c = 0.01$.

The CD channel resembles the Uniform distribution. The FFT and SVD compression methods belong to double-sided exponential distributions, thus they are more related to the Laplace distribution than to the Gaussian distribution.

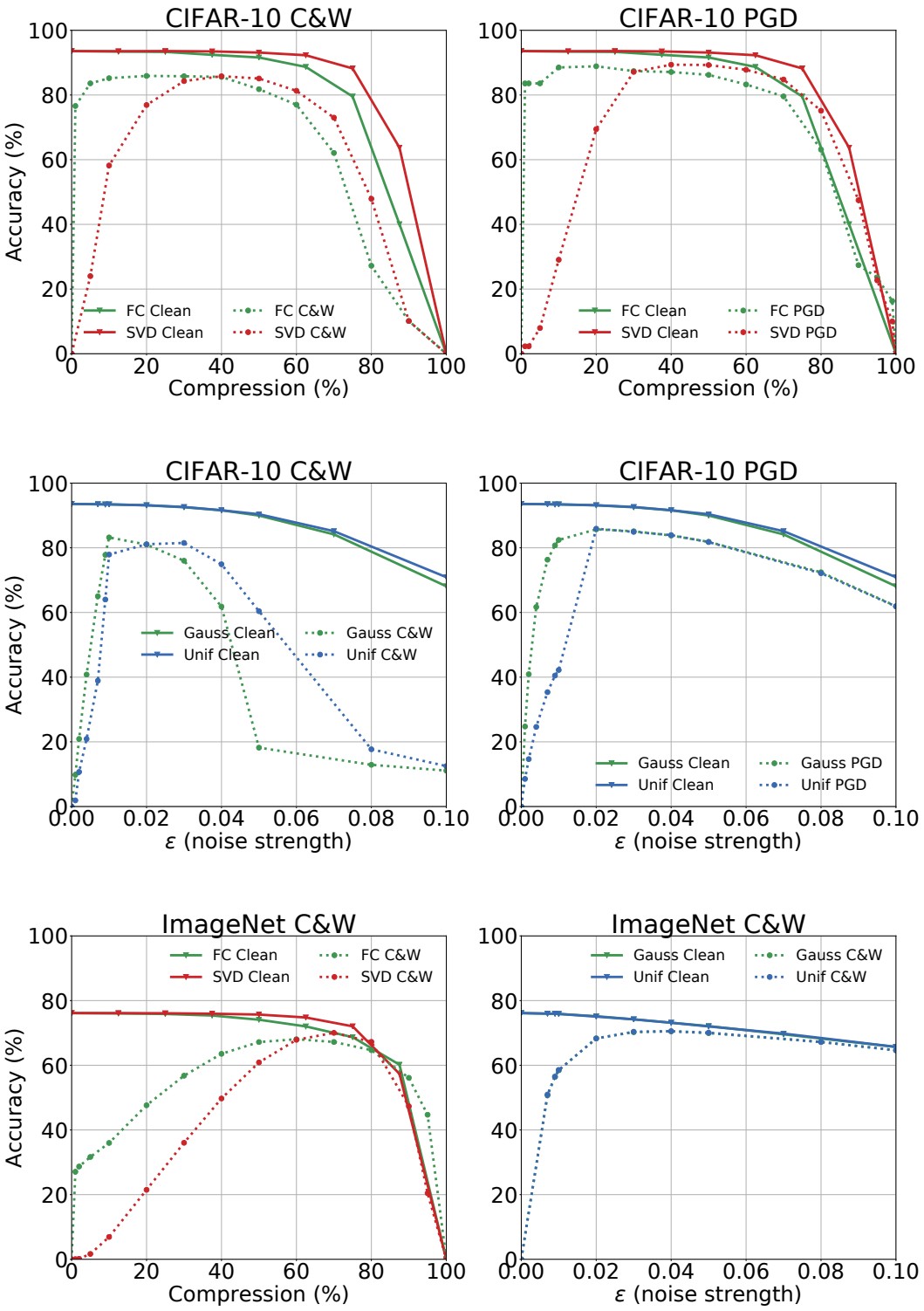

Figure 8: Comparison of accuracy of the perturbation channels on clean and adversarial images. We pass either clean or adversarial images through the channels and measure their accuracy. The accuracy on clean inputs gives us an upper bound for the accuracy on the adversarial examples. The channels can be tuned based on their accuracy after different attacks.

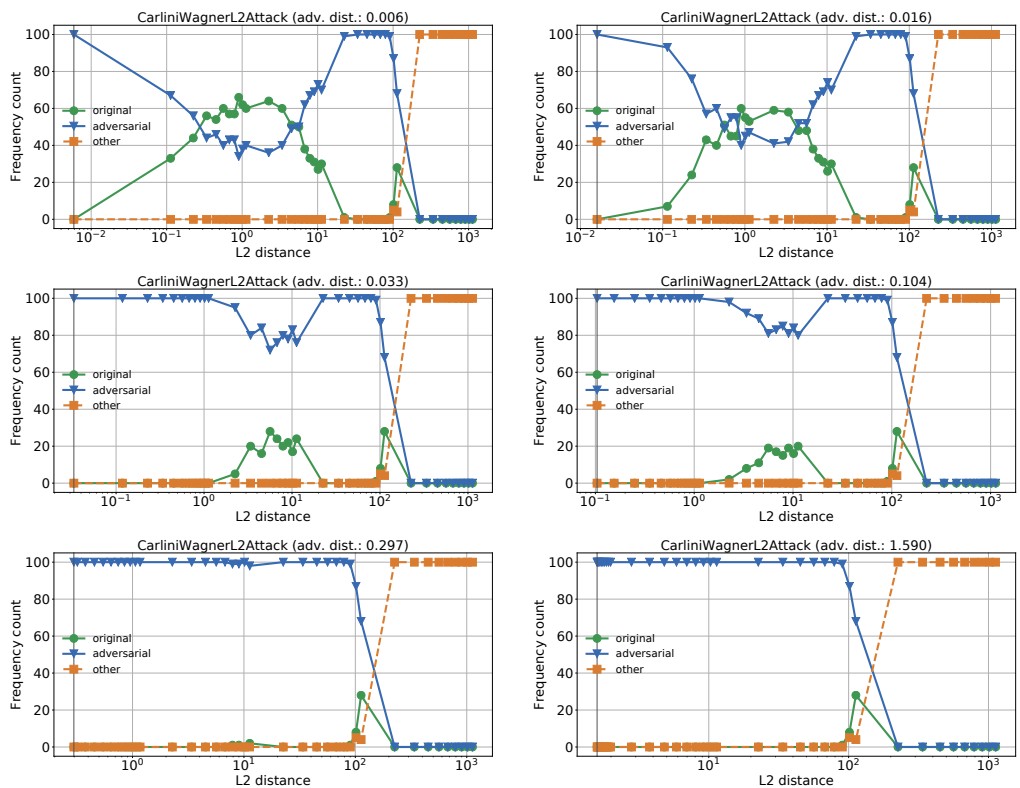

Figure 9: Frequency of model predictions for original, adversarial, and other classes as we increase the distance from an adversarial example using Gaussian noise.

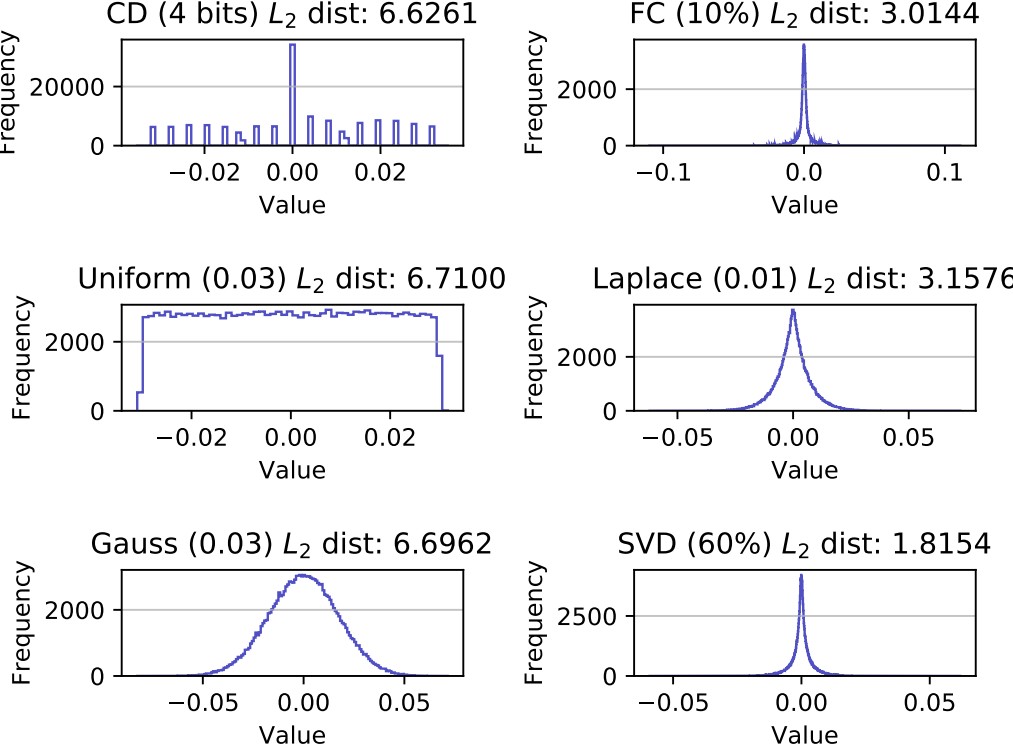

Figure 10: Distribution of *deltas* for imprecise channels.

### C.6 VISUALIZATIONS OF ATTACKS AND IMPRECISE CHANNELS

Figure 11 presents a sample image from ImageNet for the Carlini-Wagner L2 attack.

Figure 12 shows the effect of the FC (frequency-based imprecise channel) in both spatial and Fourier domains.

### C.7 MULTIPLE TRIALS FOR STOCHASTIC CHANNELS

In addition to the robustness to non-adaptive attacks, another compelling reason to use a stochastic channel (like Uniform noise) as a defense is that it can be run repeatedly in a number of random trials. We show that doing so improves the efficacy of the defense. We randomly choose 1000 images from the CIFAR-10 test set. For each of these images, we generate an adversarial attack. We then pass each image through the same stochastic channel multiple times. We take the most frequent prediction. Figure 13 illustrates the results.

Only 16 trials are needed to get a relatively strong defense. We argue that this result is significant. Randomized defenses are difficult to attack. The attacker cannot anticipate which particular perturbation to the model will happen. The downside is a potential of erratic predictions. We show that a relatively small number of trials can greatly reduce this noise. Furthermore, the expense of running multiple trials of a randomized defense is small relative to the expense of synthesizing an attack in the first place.

### C.8 WHITE-BOX ADAPTIVE ATTACK

We consider the problem setting when the adversary knows the defense method (i.e., has full knowledge of $C(x)$). We use the strategy described in He et al. (2017) to construct attacks for each case. Not surprisingly, deterministic channels (CD, FC, and SVD) are easy for an adversary to fool when they are known. However, such attacks incur higher distortion (distance to the original image)

when compared to attacks against unprotected networks. Intuitively, when the channel is deterministic and known, the adversary can account for the error introduced by the channel. Table 5 illustrates the results.

Table 5: The distortion and accuracy for the adaptive setting where the adversary knows the defense method. We report top-1 class, use 100 image samples, run 1000 iterations of C&W attack, $\epsilon = 0.04$, apply a single random noise injection.

| **CIFAR-10** | $L_2$ Distortion | Acc. (%) | | **ImageNet** | $L_2$ Distortion | Acc. (%) |
|---|---|---|---|---|---|---|
| Iden | 0.17 | 0 | | Iden | 0.28 | 0 |
| CD (b=5) | 0.5 | 0 | | CD (b=5) | 3.6 | 0 |
| FC (c=30) | 0.74 | 0 | | FC (c=30) | 3.64 | 0 |

The stochastic channel is harder to attack with a gradient-based adaptive methods. Unlike for deterministic compression, the adversary cannot anticipate the particular error pattern. We build an adaptive attack against the additive uniform noise channel. Our strategy is to send an output from the adversarial algorithm through the stochastic channel and the network at least as many times as set in the defense. We mark the attack as successful if the most frequent output label is different from the ground truth. The more passes through the noisy channel we optimize for, the stronger the attack. Furthermore, we run many iterations of the attack to decrease the $L_2$ distortion. An attack that always evades the noise injection defense is more difficult to generate because of randomization. The other randomized approach was introduced in dropout (Feinman et al., 2017). The attacks against randomized defenses require optimization of complex loss functions, incur higher distortion, and the attacks are not fully successful (Carlini & Wagner, 2017).

In the Figure 14, we present the result of running attacks and defenses on CIFAR-10 data with single and many iterations. The defense with many trials can be drawn to 0% accuracy, however, the defense not fully optimized by the adversary (single noise injection) can result in about 40% or higher accuracy.

## C.9 HYBRID APPROACHES

A natural thought is whether these defenses can be made more effective by combining them. This is an approach that has been applied, for example, in random discretization (Zhang & Liang, 2019). Our experiments contrast with the previous work and suggest that there is little benefit to hybrid approaches. We present an illustrative example for pair-wise combinations of Frequency Compression (FC), Color Depth (CD) compression, and Uniform (Unif) noise. Other combinations are possible but for brevity, we exclude them from the manuscript; we found no to very small benefit to combining for properly tuned channels. We show the recovery rate, i.e. a fraction of original labels recovered. Table C.9 presents results for the L2 Carlini-Wagner attack on the MNIST, the whole test set of CIFAR-10, and the whole development set of ImageNet.

| | CD (bits) | FC (%) | Unif ($\epsilon$) | CD+FC | FC+CD | CD+Unif | FC+Unif |
|---|---|---|---|---|---|---|---|
| MNIST | 100 | 95 | 100 | 100 | 100 | 100 | 100 |
| CIFAR-10 | 84.4(4) | 85.2(20) | 83.4(0.03) | 86.0 | 83.6 | 85.18 | 86.45 |
| ImageNet | 80.9(4) | 78.7(50) | 80.3(0.03) | 78.6 | 77.5 | 73.15 | 76.13 |

We believe these results suggest that the noisy channels do not exploit anything inherent to the images. Simply the addition of noise (through reconstruction error) is the mechanism for robustness. Composing two different schemes usually increases this noise.

## C.10 TRANSFERABILITY OF THE ADVERSARIAL IMAGES

We present more results on the transferability of adversarial examples between different channels in Table 1 and Table 7).

Table 6: Transferability of the adversarial images that extend results from Table 1 for ImageNet dataset. We use 30% FC compression, 50% SVD compression, 4 bit values in CD, 0.03 noise level for Laplace, and 0.04 noise level for the Gauss and Uniform channels. We use 3000 images from the ImageNet-10 validation set and 100 attack iterations.

| A \ D | FC | CD | SVD | Gauss | Uniform | Laplace |
|---|---|---|---|---|---|---|
| FC | 0.10 | 75.50 | 75.83 | 77.04 | 77.49 | 76.29 |
| CD | 0.17 | 1.16 | 6.77 | 62.60 | 62.04 | 65.46 |
| SVD | 12.02 | 72.33 | 0.46 | 72.79 | 72.52 | 73.09 |
| Gauss | 0.57 | 26.67 | 6.67 | 58.68 | 58.62 | 64.95 |
| Uniform | 0.50 | 26.71 | 6.99 | 58.48 | 59.06 | 64.59 |
| Laplace | 0.33 | 18.59 | 4.16 | 29.76 | 29.84 | 50.00 |

Table 7: Transferability of the adversarial images. The results are presented similarly to Figure 1 but for different parameters. We use 50% FC compression, 50% SVD compression, 4 bit values in CD, 0.03 noise level for Laplace, and 0.04 noise level for the Gauss and Uniform channels. We use 3000 images from the CIFAR-10 validation set and 1000 attack iterations.

| A \ D | FC | CD | SVD | Gauss | Uniform | Laplace |
|---|---|---|---|---|---|---|
| FC | 0.19 | 79.00 | 83.73 | 79.19 | 79.38 | 76.70 |
| CD | 6.74 | 0.93 | 47.57 | 62.98 | 63.04 | 65.22 |
| SVD | 78.89 | 47.04 | 0.50 | 59.85 | 63.42 | 67.94 |
| Gauss | 4.77 | 38.52 | 36.95 | 51.36 | 51.27 | 52.61 |
| Uniform | 4.65 | 38.14 | 36.35 | 50.14 | 51.08 | 53.21 |
| Laplace | 46.75 | 22.22 | 22.68 | 34.03 | 33.93 | 46.39 |

## C.11   GRADIENT-BASED ANALYSIS

We run the experiment on the ImageNet dataset. We analyze only the clean images that were classified correctly.

We present how the gradient of the loss w.r.t. the input image changes for the correct class as we add the Gaussian noise to the original image in Figure 15. The norm of the gradient smoothly increases.

In Figure 16, we start from an adversarial image found with the default C&W attack from the foolbox library. Then, we systematically add Gaussian noise to the adversarial image and collect data on the norm of gradients for the original and adversarial classes. The norm of the gradients for the adversarial class increases while the norm of the gradients for the original class decreases. We cross the decision boundary to the correct class very early and recover the correct labels for images. Then, as we add more and more Gaussian noise, the predictions of the classifier become random and the norm of the gradients converge to a single value.

In Figure 17, we plot the gradients also for a random class. We observe that for an untargeted attack, the gradients for the original and adversarial classes are larger than for the other classes. The targeted attack decreases the loss for the target class and the gradients for the adversarial classes are lower when compared with gradients from the untargeted attacks, so fewer images can be recovered in the former case. The targeted attack causes a smaller increase of the norms of gradients for the original class than the untargted attack. However, it is still higher than for a random class.

## C.12   HESSIAN-BASED ANALYSIS

We present the Hessian spectrum in Figure 18 for top 20 eigenvalues and in Figure 19 the distribution of top eigenvalues of Hessians on ImageNet. For the adversarial images, the eigenvalues are clearly higher in both cases, which indicates higher instability and proclivity to prediction changes.

## C.13 DETAILS ON THE EXPERIMENTAL SETUP

We use the foolbox library (Rauber et al., 2017) in our experiments. We borrowed the nomenclature used for the attacks from the library. For example, the name for the attack initially proposed in Szegedy et al. (2014) and extended in Tabacof & Valle (2015) is LBFGS. In most of our experiments, we also use the default foolbox parameters for the attacks. For example, for PGD the initial limit on the perturbation size epsilon is set to 0.3, step size to 0.01, default number of iterations is 40. For Carlini & Wagner, we set maximum number of iterations to 1000, learning rate to 0.005, initial value of the constant $c$ to 0.01. Note that for the Carlini & Wagner attack presented in 2, we use the $c$ parameter as described in (Carlini & Wagner, 2017) and the code from Liu et al. (2018). For the LBFGS attack, we use the epsilon parameter set to 0.00001 and up to 150 iterations. For the FGSM attack, different epsilons starting from 50 and up to 1000 are tried until the adversarial image is found. For the BIM $L_1$ attack, we set epsilon to 0.3, step size to 0.05, and number of iterations to 10.

# D  ADDITIONAL EXPERIMENTS FOR BLACK-BOX ATTACKS

As a black box attack, we define an attack that does not need the knowledge about the gradient or the model.

## D.1  DECISION-BASED ATTACKS

The attacks require neither gradients nor probabilities. They operate directly on the images.

### D.1.1  ROBUSTNESS TO UNIFORM AND GAUSSIAN NOISE

We evaluate the robustness of band-limited CNNs. Specifically, models trained with more compression discard part of the noise by removing the high frequency Fourier coefficients (FC channel). In Figure 20, we show the test accuracy for input images perturbed with different levels of uniform and Gaussian noise, which is controlled systematically by the sigma parameter, fed into models trained with different compression levels (i.e., 0%, 50%, or 85%) and methods (i.e., band-limited vs. RPA-based[3]). Our results demonstrate that models trained with higher compression are more robust to the inserted noise. Interestingly, band-limited CNNs also outperform the RPA-based method and under-fitted models (e.g., via early stopping), which do not exhibit the robustness to noise.

Input test images are perturbed with uniform or Gaussian noise, where the sigma parameter is changed from 0 to 1 or 0 to 2, respectively. The more band-limited model, the more robust it is to the introduced noise.

### D.1.2  CONTRAST REDUCTION ATTACK

This black-box attack gradually distorts all the pixels:

$$\text{target} = \frac{\text{max} + \text{min}}{2}$$
$$\text{perturbed} = (1 - \epsilon) * \text{image} + \epsilon * \text{target}$$

where min and max values are computed across all pixels of images in the dataset.

We can defend the attack with CD (Color Depth reduction) until certain value of epsilon, but then every pixel is perturbed in a smooth way so there are no high-frequency coefficients increased in the FFT domain of the image. The contrast reduction attack becomes a low-frequency based attack when considered in the frequency domain. Another way to defend the attack is to run a high-pass filter in the frequency domain instead of the low-pass filter.

We run the experiments for different models with CD and two band-limited models (the model with full spectra and no compression as well as model with 85% of compression - with FC layers). The CD does defend the attack to some extent and the fewer pixels per channel (the *stronger* the CD in a model), the more robust the model is against the contrast reduction attack.

Test accuracy as a function of the contrast reduction attack for ResNet-18 on CIFAR-10 (after 350 epochs) is plotted in Figure 20. We control the strength of the attack with parameter epsilon that is changed systematically from 0.0 to 1.0. We use the whole test set for CIFAR-10. R denotes the number of values used per channel (e.g., R=32 means that we use 32 values instead of standard 256).

### D.1.3  MULTIPLE PIXELS ATTACK

The foolbox library supports a single pixel attack, where a given pixel is set to white or black[4]. A certain number of pixels (e.g., 1000) is chosen and each of them is checked separately if it can lead to the misclassification of the image. The natural extension is to increase the number of pixels to be perturbed, in case where the single pixel attack does not succeed. We present results for the multiple pixel attack in Figure 20.

---

[3]The Reduced Precision Arithmetic, where operations on 16 bit floats are used instead of on 32 or 64 bit float numbers.

[4]The single and multiple pixels attacks are categorized as decision-based attacks since they treat the model as an oracle and require access neither to gradients nor probabilities.

## D.2    SPATIAL-BASED ATTACKS

Spatial attacks apply adversarial rotations and translations that can be easily added to the data augmentation during training. However, these attacks are defended neither by removing the high frequency coefficients nor by quantization (CD) . We separately apply rotation by changing its angle from 0 to 20 degrees and do the translations within a horizontal and vertical limit of shifted pixels (Figure 20).

## D.3    SCORE-BASED ATTACKS

The score based attack require access to the model predictions and its probabilities (the inputs to the softmax) or the logits to estimate the gradients.

### D.3.1    LOCAL SEARCH ATTACK

The local search attack estimates the sensitivity of individual pixels by applying extreme perturbations and observing the effect on the probability of the correct class. Next, it perturbs the pixels to which the model is most sensitive. The procedure is repeated until the image is misclassified, searching for additional critical pixels in the neighborhood of previously found ones. We run the experiments for the attack on 100 test images from CIFAR-10, since the attack is relatively slow (Figure  20).

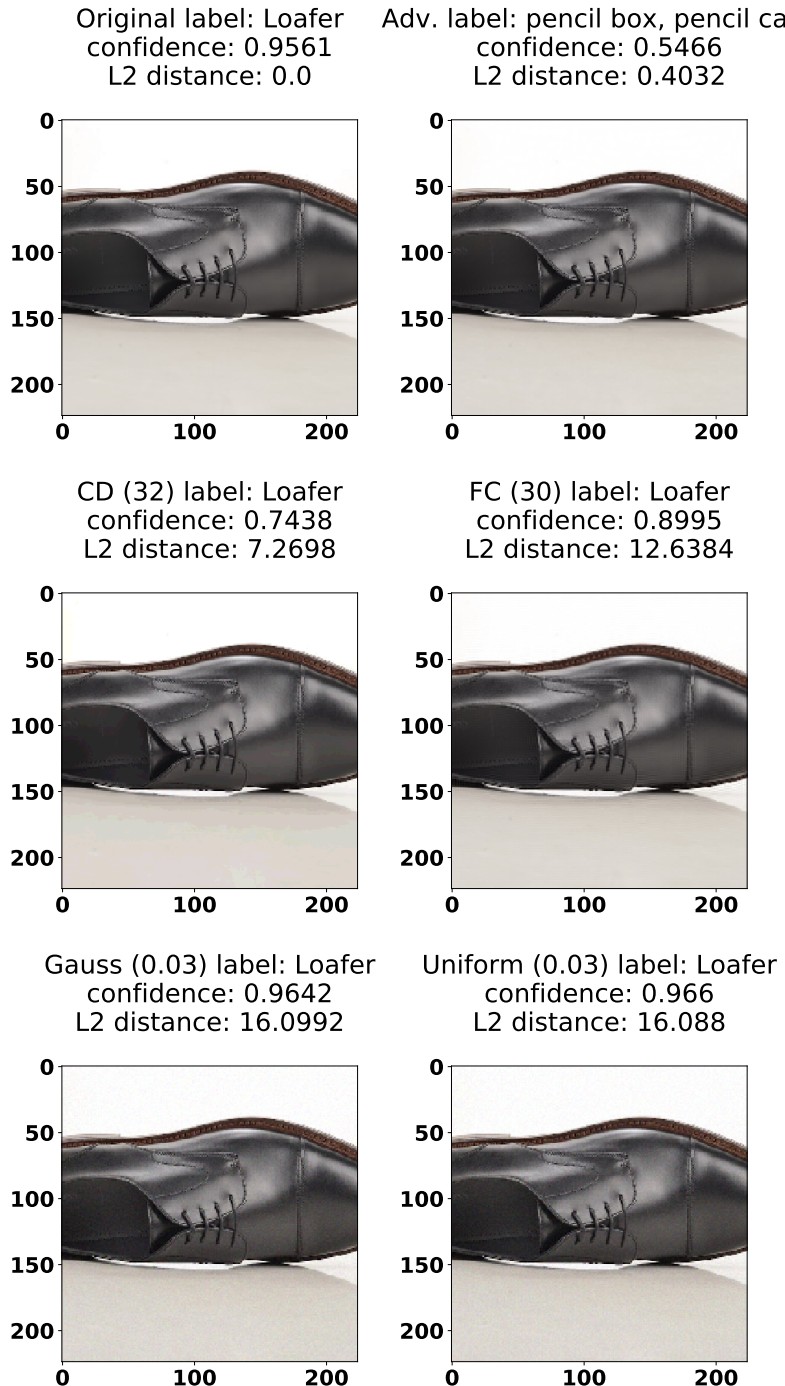

Figure 11: We plot a sample image from the ImageNet dataset in its original state, after adversarial (white-box, non-adaptive) attack, and then after recovery via imprecise channels: CD (color depth reduction with 32 bits), FC (30% compression in the frequency domain), Gaussian, and uniform noise ($\epsilon = 0.03$). The order is from left to right, and from top to bottom.

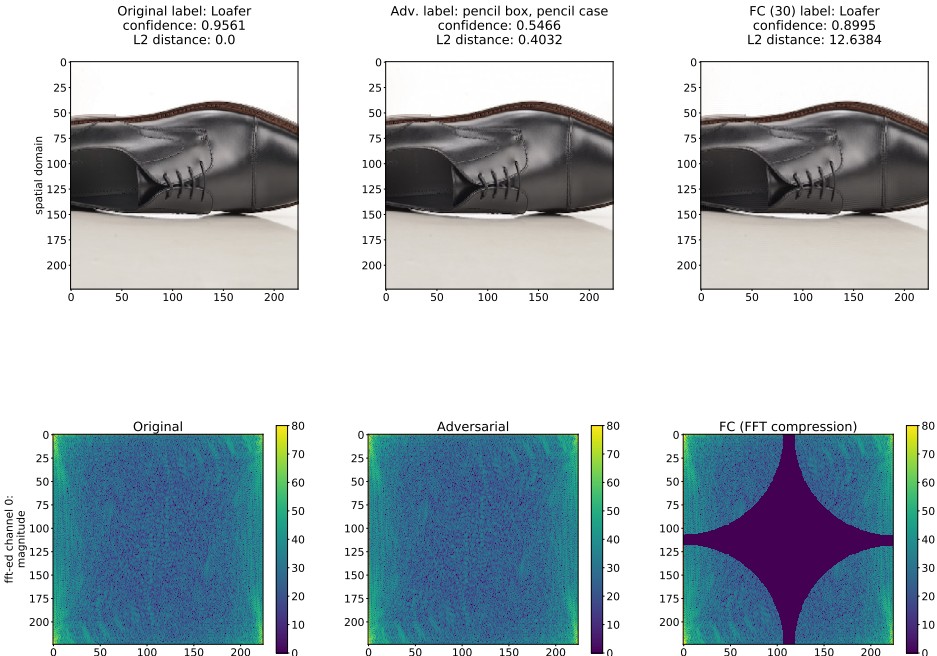

Figure 12: The original image from the ImageNet dataset, its adversarial example, and the state of the image after recovery from the attack via the 30% compressed Fourier Channel (FC). The heat maps of magnitudes of Fourier coefficients are presented in a logarithmic scale (dB) with linear interpolation and the max value is colored with white while the min value is colored with black. The black part of the (bottom-right) image represents the removed high-frequency coefficients. The Fourier-ed representation is plotted for a single (0-th) channel. The lowest frequency coefficients are placed in the corners of the FFT maps (with the DC component in the top-left corner).

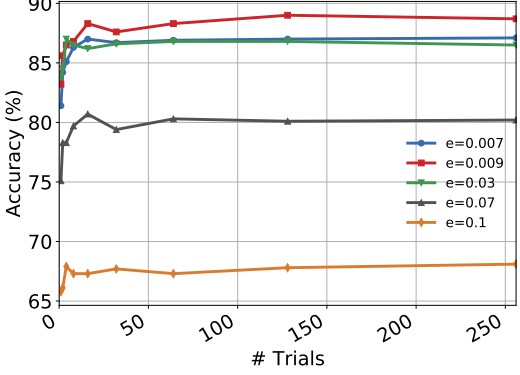

Figure 13: For the CIFAR-10 dataset, we run multiple trials of the uniform noise channel and take the most frequent prediction. We further test multiple noise levels. The multiple trials improve overall accuracy for different noise levels significantly. After 128 trials for the best setting we are within 3% of the overall model accuracy (of about 93.5%).

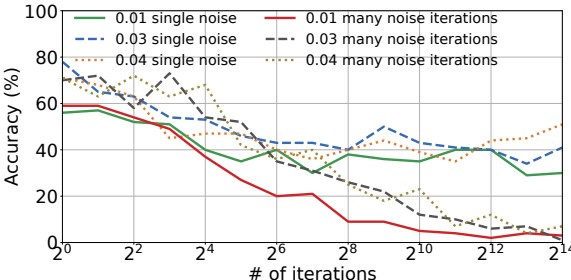

Figure 14: For the CIFAR-10 dataset, we run multiple trials of the uniform noise channel and take the most frequent prediction in the defense (many noise iterations). We also run just a single noise injection and return the predicated label. The attacker runs the same number of many uniform trials as the defender. The experiment is run on 100 images, with 100 C&W $L_2$ attack iterations.

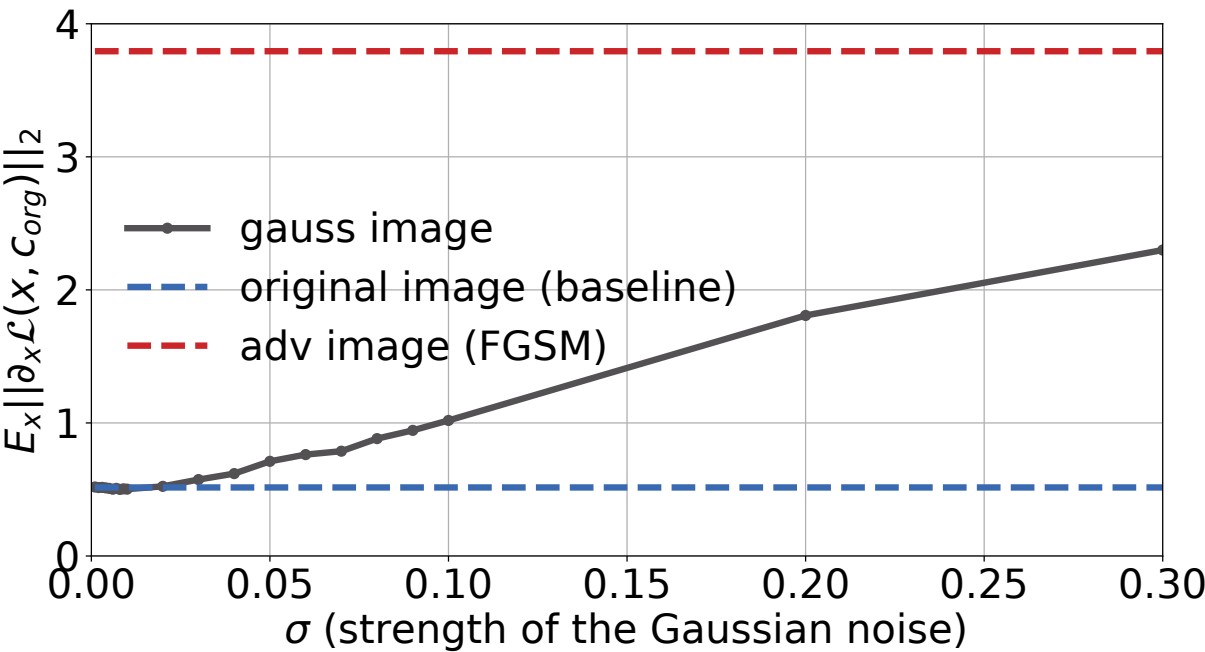

Figure 15: The changes in the $L_2$ norm of the gradient of the loss w.r.t. the input image $x$ for the correct class $c_{org}$ as we add Gaussian noise to the original image. The experiment is run on 1000 images from the ImageNet dataset.

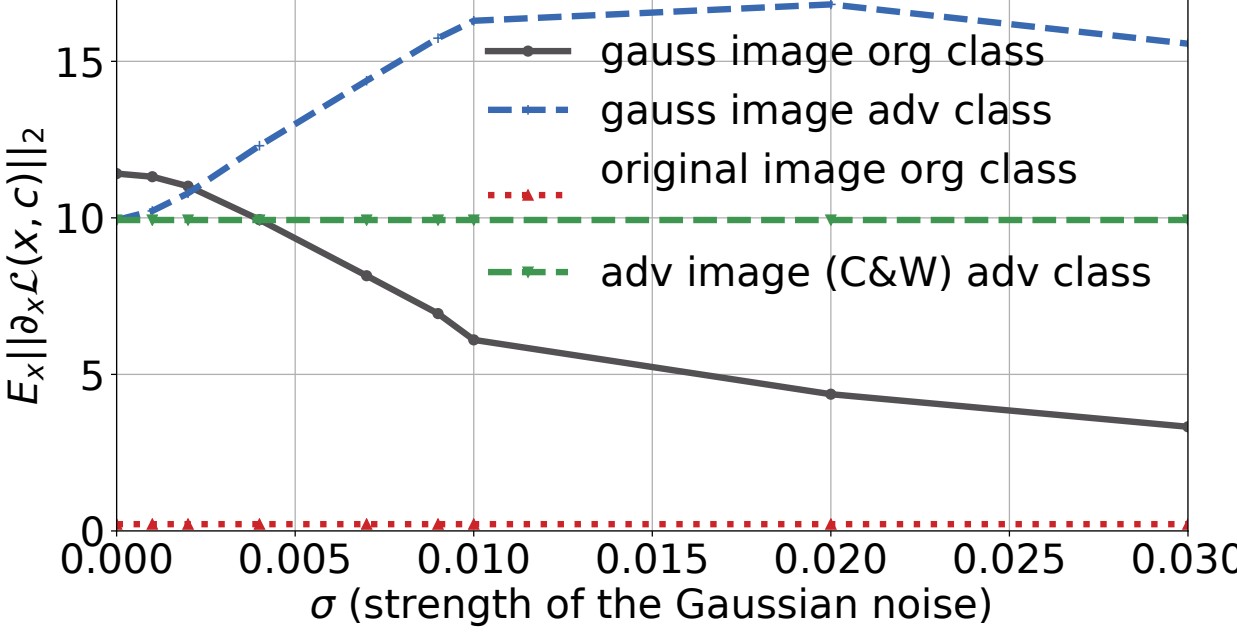

Figure 16: The changes in the $L_2$ norm of the gradient for the correct class $c_{org}$ and the adversarial class $c_{adv}$ as we add Gaussian noise to the adversarial image generated with C&W $L_2$ attack (we label such an image as a *gauss image* $x$). The experiment is run on 1000 images from the CIFAR-10 dataset.

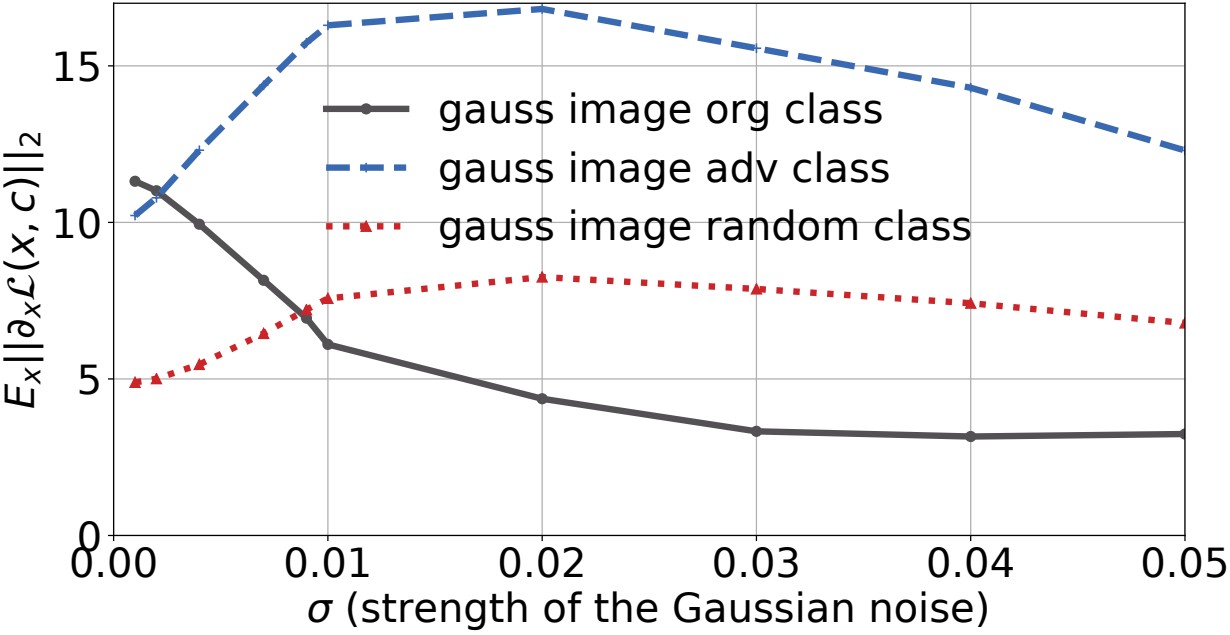

Figure 17: The changes in the $L_2$ norm of the gradient of the loss for the correct class $c_{org}$, the adversarial class $c_{adv}$, and a random class $c_{ran}$ as we add Gaussian noise to the adversarial image generated with C&W $L_2$ attack (we label such an image as a *gauss image x*). The experiment is run on 1000 images from the CIFAR-10 dataset.

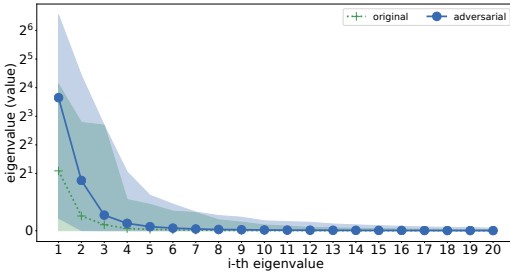

Figure 18: The spectrum of the Hessian with respect to the original and adversarial inputs. We use the symmetric log scale on the y axis and plot the min and max ranges of the eigenvalues for 100 images from the ImageNet dataset trained the ResNet-50 architecture.

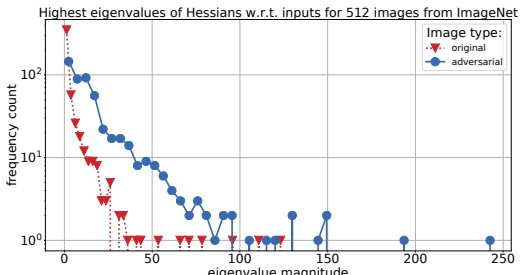

Figure 19: The top eigenvalues of the Hessians on ImageNet using ResNet-50.

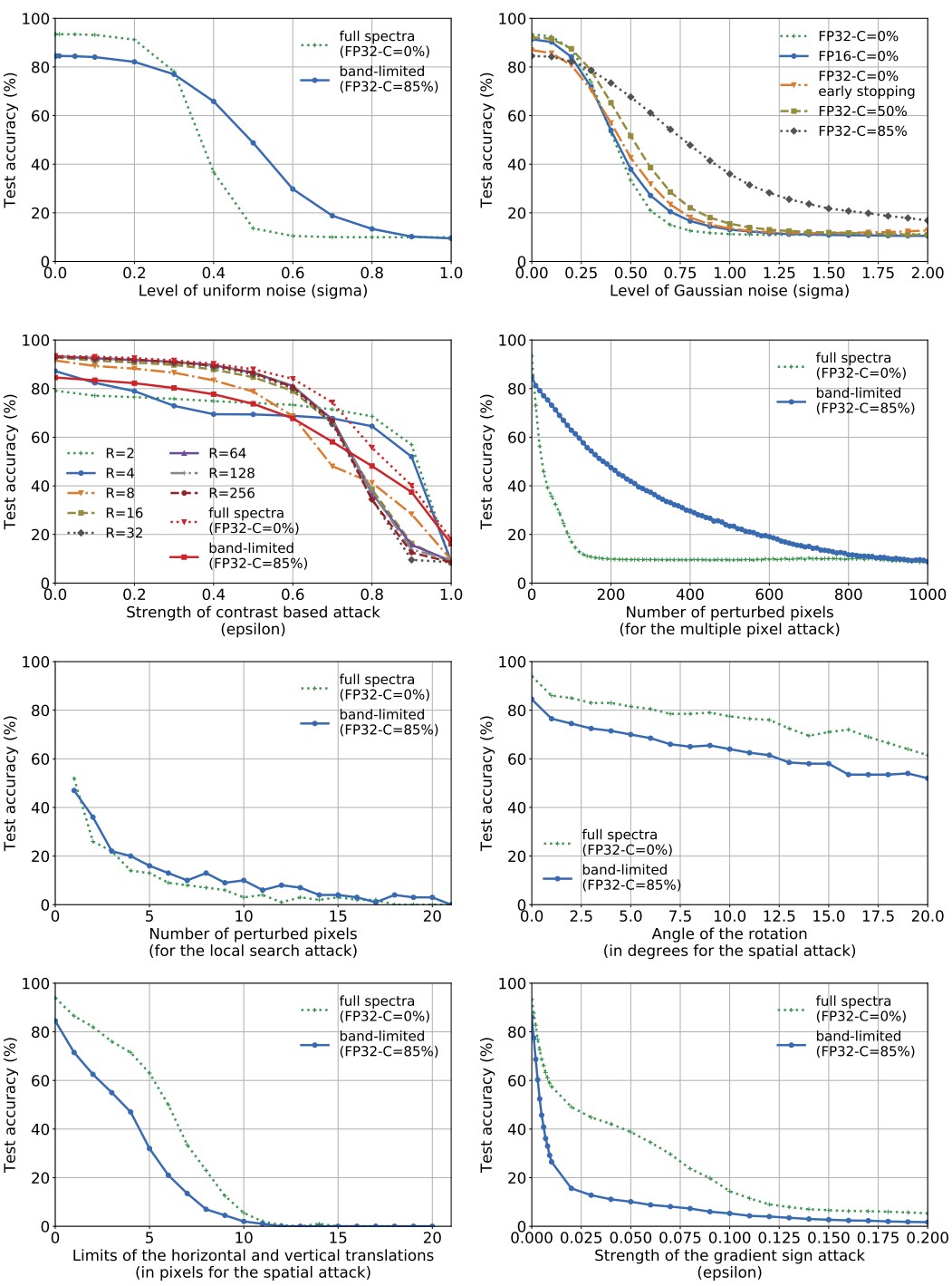

Figure 20: Test accuracy as a function of the strenghts of the attacks for ResNet-18 on CIFAR-10.

