# OpenReview forum: "A Perturbation Analysis of Input Transformations for Adversarial Attacks"
_ICLR.cc/2020/Conference — Reject_

### Official Review · AnonReviewer3 · 2019-10-22
**Official Blind Review #3**

**Rating:** 6

**Review:**

The work is concerned with a very interesting question: "robustness of the adversarial attacks" and discusses the following: First, what are different ways of destabilizing a given adversarial attack (different perturbations) and how much each of them are effective. Secondly, what makes adversarial images particularly non-robust compared to natural images and is there empirical evidence for it? Thirdly, is it possible for an attacker to use this knowledge and use a rather universal model of perturbations to make its adversarial examples robust against such (deterministic or stochastic) perturbations?

The paper is quite well-justified and addresses a very interesting question by unifying the existing results of the literature in one place. First of all, comparing the accuracy-robustness trade-off of different methods are useful.  The idea of modeling perturbations as a general distorted communication channel to have a unification of these methods is very useful. The perturbation analyses along with the results in Figures 2, 3 are novel and can be useful for further theoretical investigation of the source of instability of adversarial images. To the best of my knowledge, the transferability of adaptive attacks against noisy channel defense methods has not been discussed in the literature. The paper is well-organized and well written (except for Section 5 both in sense of the writing and where it appears in the paper).

A few questions and suggestions:

* The discussion of results in Figure2 should be more extensive and more clear.

* Why do you think the L2 of the gradient for adv images has such high variance for both orig and adv class? Is there a relationship between the magnitude and how effective is a perturbation-based defense?

* It should be made clear in the text that methods like randomized smoothing are designed for the purpose of 'certified' accuracy and being robust against attack sizes more than the certification threshold was not part of the method's contributions.

* The use of the term "compression" in Section 3 is not entirely correct as it is not "technically" correct to say that compression is what is necessarily happening in a general case of a deterministic C(x).

* Fig 1 should be larger.

**Experience Assessment:**

I have read many papers in this area.

**Review Assessment: Checking Correctness Of Derivations And Theory:**

I carefully checked the derivations and theory.

**Review Assessment: Checking Correctness Of Experiments:**

I carefully checked the experiments.

**Review Assessment: Thoroughness In Paper Reading:**

I read the paper thoroughly.

---

> ### Author Response · Authors · 2019-11-07
> **Response to Review#3**
>
> Thank you very much for the positive and constructive feedback.
>
> We moved Section 5 before Section 4 and re-wrote it.
>
> We extended the description of the gradient-based analysis. We focused on the fact that adversarial examples have larger gradient norms than the natural inputs.
>
> We corrected the description of randomized smoothing.
>
> For the imprecise channels, we focused on compression and randomization techniques. We added more explanation of the FFT (denoted FC) and SVD compression techniques in Appendix B. Indeed, compression is not what is necessarily happening in a general case of a deterministic C(x) and we also included a very simple other deterministic C(x) that reduces brightness of an image by subtracting an arbitrary value from each pixel (it used to be Section 5 but was moved to Section 4, and currently is presented in Fig 2).
>
> The extension of results from Fig 1 is shown in larger Fig 7 (in the Appendix).

---

> > ### Comment · AnonReviewer3 · 2019-11-13
> > **Thanks**
> >
> > Thank you very much for addressing the questions. Although I agree with the other reviewers' novelty concerns, due to the importance of the studied problem my score will remain unchanged.
> > PS: The font in Fig 1 should be larger at least :) Also there is a reference typo on page 12.

---

> > > ### Author Response · Authors · 2019-11-13
> > > **Font & reference**
> > >
> > > Thank you very much, we appreciate your feedback. We increased the font in Fig 1 and corrected the reference typo.

---

### Official Review · AnonReviewer2 · 2019-10-22
**Official Blind Review #2**

**Rating:** 3

**Review:**

The paper studies the robustness of adversarial attacks to transformations of their output. Specifically, for standard methods for crafting adversarial examples, the authors evaluate whether the crafted examples remain adversarial under (stochastic or deterministic) transformations such as Gaussian noise and SVD compression. The authors argue that different transformations have a similar impact on the perturbed inputs. Finally, they argue that the reason why these transformations can sometimes recover the correct label is due to the loss being more unstable at these points (from a first- and second-order pespective).

From a conceptual point of view, I did not find the paper particularly impactful. In particular, I can identify three claimed contributions:

a) The L2 distortion introduced by these transformation might have more impact than the specific transformation used (figure 1).
Firstly, I would argue that this effect is most prominent for small L2 distortions (distortions larger than 5/40 do exhibit noticeable differences). Secondly, I am not sure why one would expect these transformations to have fundamentally different impact. The adversarial attacks considered manipulate the input using first-order methods in input space. Since the transformations are model- and data-agnostic it seems expected that the primary mechanism behind their effect is the pixel-wise distortion of the image.

b) Adversarial perturbations that are robust to one type of transformation tend to also be robust to other transformations (table 2).
This is probably the most interesting observation of the paper, indicating that attackers can bypass several of these transformation defenses by only aiming to be robust to a subset of them. At the same time, I am not sure what the impact of this observation is given that we already know how to bypass most of these defenses anyway.

c) The instability of adversarial perturbations can be explained by a larger gradient norm and Hessian spectrum (figure 2, 3).
First of all, I do not understand how this is considered a potential explanation of empirical behavior. If gradient norm is indicative of instability, then _natural_ images would be unstable (since the gradient wrt the wrong class is large). Furthermore, instability does not explain why adding noise leads to the _correct_ class as opposed to a random class. Perhaps most importantly, from what I understand (also skimming the code), Figure 2 plots the gradient of the _cross-entropy loss_ with respect to each class. However, the norm of the gradient is directly affected by the softmax probability of each class. Hence, for natural images, the probability of the correct class is high leading to a small norm while for the adversarial class it is low leading to large norm. Based on this reasoning, this plot does convey information about the classifier's stability but rather about the softmax probabilites assigned to each class.

In summary, after reading the paper, I am not sure how our understanding of transformation robustness has changed or what insight we have gained that will help us design future attacks and defenses (especially given prior work on how most of these defenses can be bypassed). I thus recommend rejection.

Comments to authors:
-- Many of the attack details are missing (what is the epsilon allowed for PGD? what does the "c" parameter correspond to for CW attacks?) which prevented me from fully comprehending the experimental results.
-- LBFGS is not a particular attack, it is a general optimization algorithm, https://en.wikipedia.org/wiki/Limited-memory_BFGS
-- The fact that adversarial points are close to correctly classified points does not mean that a small amount of random noise will change the classifier prediction. In high dimension, the distance to a hyperplane can be epsilon but moving in a random direction requires movement of epsilon * sqrt(number of dimensions) .

==============

UPDATE: I appreciate the authors' response. As stated in my original review, I do recognize the performance of a non-adaptive adversary as a contribution. However, I still don't think that the impact of this finding is significant enough for publication at ICLR. I hence keep my original score.

Response to specific points:
-- Unfortunately, the authors did not address my concerns about the gradient of the loss. The experiments and reasoning of the paper should be sufficient without citing other work in the author response. Moreover, it is unclear if the papers referenced in the response are showing fundamental properties of adversarial examples or are specific to the attacks considered.  For instance, there has been work challenging these papers (https://arxiv.org/abs/1907.12138).

-- As mentioned in my original review, measuring the norm of the gradient of the softmax loss will inevitably take into account the predicted probability of each class. Hence in order to truly measure sensitivity it would be important to work with the logits instead of the softmax probabilities.



**Experience Assessment:**

I have published in this field for several years.

**Review Assessment: Checking Correctness Of Derivations And Theory:**

I carefully checked the derivations and theory.

**Review Assessment: Checking Correctness Of Experiments:**

I carefully checked the experiments.

**Review Assessment: Thoroughness In Paper Reading:**

I read the paper thoroughly.

---

> ### Author Response · Authors · 2019-11-12
> **Response to Review #2**
>
> We would like to thank the reviewer for the insightful comments.
> Ad.a) The interesting observation is that the content-preserving transformations (FFT or SVD compression) have a very similar recovery rate and the maximum accuracy achieved after attacks to the content-oblivious transformations (Gaussian, Uniform, or Laplace noise).
> Ad.b) While many of the defenses presented in the paper were bypassed, it was done in the adaptive setting, where the attacker has *full* knowledge of the defense strategy. Imagine an attacker that does not know the particular defense but anticipates that some form of perturbation is used. Our study of unification explicitly shows the transferability of attacks. This property is relevant because the attacker can devise an attack on Laplace perturbations and it will be effective on other defenses as well.
> Ad.c) The work by Simon-Gabriel et al. (2019) https://arxiv.org/abs/1802.01421 also evaluates the norm of gradients of the network output with respect to its inputs. They show that the adversarial examples are primarily caused by large gradients of the classifier as captured via the induced loss. At first-order approximation, an $\epsilon$-sized $L_2$ *untargeted* adversarial attack increases the loss $L$ at point $x$ by $\epsilon ||\partial_x L(x, c=org)||_2$, where $c$ is the original class. Analogously for the *targeted* attack, an $\epsilon$-sized $L_2$ targeted adversarial attack decreases the loss $L$ at point $x$ by $\epsilon||\partial_x L(x, c=adv)||_2$, where $c$ is the adversarial class. Thus, the lower the gradient of the loss w.r.t. input and both *original* and *adversarial* classes, the harder it is to attack the image and it is more stable.
> - From another perspective, the norm of the input gradient with respect to the predicted class is not enough on its own to understand the sensitivity of adversarial examples. Consider a 3-class example originally predicted with class confidences [Class0=90%, Class1=5%, Class2=5%]. A first-order analysis of the gradients will show that the input gradients for classes 1, 2 are large, but for class 0 it is very small. This means that random perturbations can potentially increase the confidence levels of the incorrect classes but not necessarily decrease the confidence for the predicted class. Our results suggest that both are necessary for recovery, you need a perturbation that increases the confidence in the original class while simultaneously reducing the confidence of the adversarial class. Hence, we present the results in a scatter plot for original and adversarial classes and a regime (region of the plot) in which perturbations tend to recover the original class. When taken together, it is not only enough that the gradient with respect to the not-predicted classes is large, but also that the gradient with respect to the predicted class is also relatively high. We believe that this results in perturbations that generally decrease the confidence in the predicted class and raise the confidence in the other classes.
> - We added more details about the attacks (in Appendix C.13). Regarding the CW attack, we use the c parameter as described in https://arxiv.org/abs/1608.04644
> - We use the foolbox library https://arxiv.org/abs/1707.04131 and the name of the attack described in https://arxiv.org/abs/1510.05328 is LBFGS. We borrowed the nomenclature from this library. In most of our experiments, we also use the default foolbox parameters for the attacks. For example, for PGD the initial limit on the perturbation size epsilon is set to 0.3.
> - This paper https://arxiv.org/abs/1910.07629 explains that the minimum distance from a natural input to a decision boundary of any incorrect class is small, but the density of directions that can lead to the decision boundary within a short distance is also very low. Thus, random perturbations of the input should not lead to changes in the predicted label. On the other hand, the adversarial directions can be found, for example, with gradient methods (e.g. PGD) and they require very few gradient steps to reach the decision boundary of any random target class.
> - The work by Roth et al. (2019) https://arxiv.org/abs/1902.04818 shows that adversarial examples are much closer to the unperturbed sample than to any other neighbor. The probability of the correct class increases faster than the probability of the highest other class when adding noise with a small to intermediate magnitude to the adversarial example. In our section 4.3, for the (non-adaptive) adversarial examples that are very close to the original image, the window of recovery (which indicates which levels of the random noise can recover the correct label) is relatively wide, however, as we increase the distance of the adversarial image from the original image, the window of recovery shrinks and finally we are not able to recover the correct label but have to resort to other statistics as proposed in Roth et al. (2019).

---

> > ### Author Response · Authors · 2019-11-15
> > **Instability of original and adversarial examples**
> >
> > For natural images, we can indeed find an imperceptible specific adversarial perturbation to make the model misclassify the input. For such non-adaptive adversarial images, it suffices to add noise to change the classifier prediction to the correct class. Thus, finding the adversarial directions is harder than finding directions back to the correct class from an adversarial example. There is a significant difference between the adversarial and random perturbations. The brand mark of adversarial noise is that different coordinates add up, instead of statistically cancelling each other out as they do with random noise.

---

> > ### Author Response · Authors · 2019-11-15
> > **Softmax probabilities and norms of gradients**
> >
> > The softmax probabilities and norms of gradients are correlated for original images but not for adversarial examples. We run experiment for 1000 images from the CIFAR-10 dataset, the classification accuracy is 93.9%. We take into account only the correctly classified images. For each image we generate an adversarial example using default C&W attack from the foolbox library. We record the softmax probabilities and norms of the gradients for each of the 10 classes. For 99% of the original images, the lowest gradients are for the original class. Only for 1% of the adversarial examples, the lowest gradients are for the adversarial class.

---

### Official Review · AnonReviewer1 · 2019-10-23
**Official Blind Review #1**

**Rating:** 3

**Review:**


This paper studies the noise injection as defense methods against adversarial perturbations. It presents several experiments on the relationship between clean and robust accuracy. Conclusions of this study are (1-1) several defense methods have the same underlying mechanism (noise injection) and behave similarly against adversarial perturbations, (1-2) all of the defense methods can be attacked by the same black-box attack, and (1-3) the reason of the correct label recovery by noise injections might be the input instability on adversarial inputs.

This paper should be rejected because (2-1) some conclusions are not well-supported by the experiments, and (2-2) the paper fails to demonstrate the novelties and their importance of some conclusions.

Major comments:
(3-1) Experiment 1 aims to highlight the similarity between the ``perturbations defenses''. However, the "similarity" is not defined, and the arguments are mostly subjective. For example, SVD and Gaussian noise appear to have different effectiveness (Figure 1). If the similarity means the observation that large distortions decrease the accuracy, it will not be surprising because all methods should achieve the chance rate accuracy when the distortions are extreme.
(3-2) What is the point of Experiment 2? The phenomenon that strong defense methods decrease clean accuracy has been observed in many defense papers. The noise injection-based defense methods will not be exceptions. Remarking the existence of the trade-off will not be a strong contribution. The focus of the discussion should be on how to take a good balance or improve both of them.
(3-3) To my best knowledge, experimentally showing that adversarial inputs have larger gradient norms (Section 4) is novel, and it is potentially interesting. I wonder how large the gradient norms will be for randomly perturbed images.

===== UPDATE=====
Thank you for the response. After reading the response and other reviews, I mostly agree with Reviewer 2 and keep my score.

Thank you for the clarification on (3-1) in the initial review. I think it made the contribution of the paper clearer. However, as I and Reviewer 2 commented in their initial review, I did not find the observation significant.
I also thank you for the additional experiments concerning (3-3). It addressed a concern that the larger gradient norms are not specific to adv. examples and appear everywhere around the natural images. However, concerns c) raised by Reviewer 2 seems critical. It is unclear whether the instability explains the observed phenomena and whether Experiment 3.3 is enough to confirm the existence of the instability. I think the paper is better to address the concerns before publication.

**Experience Assessment:**

I have published one or two papers in this area.

**Review Assessment: Checking Correctness Of Derivations And Theory:**

I assessed the sensibility of the derivations and theory.

**Review Assessment: Checking Correctness Of Experiments:**

I assessed the sensibility of the experiments.

**Review Assessment: Thoroughness In Paper Reading:**

I read the paper at least twice and used my best judgement in assessing the paper.

---

> ### Author Response · Authors · 2019-11-12
> **Response to Review #1**
>
> We thank the reviewer for the valuable comments.
>
> - (3.1) In Figure 1, we present the whole spectrum of channel distortions. We start from each channel incurring no distortion and then systematically increase it. The main point is that all the channels can be tuned to recover very similar maximum accuracy after attacks. For example, all the channels can achieve about 85% accuracy for the CIFAR-10 dataset after the (gray-box non-adaptive) PGD or Carlini & Wagner attacks.
> - We also compare the distributions of channel perturbations. We compute the deltas by subtracting an original image from the perturbed adversarial image and plot the histograms of differences. We use the ImageNet dataset. For all the examples, the correct labels are recovered after passing the adversarial examples through the channels. We use the Carlini & Wagner attack with 1000 iterations and the initial value c is set to 0.01. The results show that the CD channel resembles the Uniform distribution. The interesting point is that the FFT and SVD compression methods belong to double-sided exponential distributions, thus they are more related to the Laplace distribution than to the Gaussian distribution (please see Appendix C.5).
> - (3.2) We moved the results for Experiment 2 (Accuracy of Perturbation Defenses on Clean Data) to the Appendix (C.1). Furthermore, we discuss how to strike the right balance between the channel distortion and the accuracy after the attacks. We present the results in the Appendix (C.3) and Figure 8.  First, we compare the accuracy of the perturbation channels on clean and adversarial images. We pass either clean or adversarial images through the channels and measure their accuracy. The accuracy on clean inputs gives us an upper bound for the accuracy on the adversarial examples. Second, we tune the channels to obtain the highest accuracy after different attacks.  We used the whole test set from CIFAR-10 and the whole validation set from ImageNet, two deterministic channels (FFT and SVD compression) and two noisy channels (Gaussian and Uniform noise), two attacks (C&W and PGD).
> - (3.3) We extended the first-order analysis and observe that the magnitude of the gradient for the original class w.r.t. the input image increases smoothly as we systematically add more Gaussian noise to the original image (please see Figure 15).
> - Similarly, we also start with an adversarial image generated using the Carlini & Wagner L2 attack. Then we add Gaussian noise and see that the gradient w.r.t. the Gaussian image for the correct glass decreases and for the adversarial class increases smoothly (please see Figure 16).
> - For the original image, the gradient w.r.t the original class is very small and indicates a stable prediction. The gradient for the adversarial class w.r.t. the adversarial image is larger, however it decreases as we increase the attack strength and the confidence of the adversarial prediction.

---

### Decision · Program_Chairs · 2019-12-19

**Decision:**

Reject

**Comment:**

This paper presents an analysis on different methods of noise injection in adversarial examples, using gaussian noise for example. There are important issues raised by reviewers 1 & 2 about some conclusions not being well supported by the experiments and the utility/importance of some conclusions. After a discussion among reviewers, as of now all 3 reviewers stand by the decision that substantial improvements, and analysis can be made in the paper. Thus, Im recommending a Rejection.